# BENCHMARKS AND CUSTOM PACKAGE FOR ELECTRICAL LOAD FORECASTING

## ABSTRACT

Load forecasting is of great significance in the power industry as it can provide a reference for subsequent tasks such as power grid dispatch, thus bringing huge economic benefits. However, there are many differences between load forecasting and traditional time series forecasting. On the one hand, the load is largely influenced by many external factors, such as temperature or calendar variables. On the other hand, load forecasting aims to minimize the cost of subsequent tasks such as power grid dispatch, rather than simply pursuing prediction accuracy. In addition, the scale of predictions (such as building-level loads and aggregated-level loads) can also significantly impact the predicted results. In this paper, we provide a comprehensive load forecasting archive, which includes load domain-specific feature engineering to help forecasting models better model load data. In addition, different from the traditional loss function which only aims for accuracy, we also provide a method to customize the loss function and link the forecasting error to requirements related to subsequent tasks (such as power grid dispatching costs) integrating it into our forecasting framework. Based on such a situation, we conducted extensive experiments on 16 forecasting methods in 11 load datasets at different levels under 11 evaluation metrics, providing a reference for researchers to compare different load forecasting models.

## 1 INTRODUCTION

Time series data are becoming ubiquitous in numerous real-world applications (Wen et al., 2022; Lai et al., 2021; Zhou et al., 2022; Wang et al., 2018). Among them, electrical load forecasting is crucial for maintaining the supply and demand balance in the power system. Thanks to the development of machine learning in recent years, various methods have been developed for load forecasting (Yildiz et al., 2017; Zhang et al., 2021). To further promote the development of this field, many power load forecasting competitions like the Global Energy Forecasting (GEF) Competition have been held over the years (Hong et al., 2014; 2016; 2019). In addition, many competitions target specific themes, like building energy management based on electricity demand and solar PV generation (Nweye et al., 2022) and the impact of COVID-19 issues on the power systems (Farrokhabadi et al., 2022).

Although many advanced load forecasting methods have emerged in the past decades, the winners of load forecasting competitions often use conventional machine learning models (like non-deep learning models). The secret to their victory lies in targeted feature engineering and adjustment of forecasting strategies, which is also the major difference between load forecasting and general time series forecasting (Sobhani et al., 2020). However, no existing benchmarks focus on those parts in load forecasting. Compared with other time series, electrical load data are greatly affected by external factors such as temperature and calendar variables, making it challenging to model the load dynamics accurately. Therefore, exploring the impact of external factors on load forecasting has always been an important research direction in this field (Aprillia et al., 2020). And temperature is considered to have a significant impact on the power load. Many researchers have focused on how to use temperature variables to assist in constructing load forecasting models (Haben et al., 2019; Sobhani et al., 2020; Liu et al., 2023). At present, the utilization of temperature variables can be roughly divided into two strategies. One is to make targeted transformations on temperature variables, which are often based on relatively simple statistical learning methods (Guan et al., 2021; Farfar & Khadir, 2019). The other one is to extract features by neural networks. Such models usually achieve better accuracy (Imani, 2021; Hafeez et al., 2020). However, the interpretability of this kind of model decreases due to the

black-box characteristic of neural networks. Nevertheless, related feature engineering also has a guiding role for neural network-based forecasting models. Currently, no large-scale experimental results have been provided to demonstrate this. Therefore, we will provide various related feature engineering in our package and discuss the impact on load forecasting models based on temperature feature engineering.

Apart from feature engineering, another difference is that the most important concern of power load forecasting models is to achieve the lowest cost instead of the best accuracy of predictions. Due to the diversity of the time series, general time series forecasting results are rarely optimized for a specific task. However, the load forecasting results will mainly be used for subsequent power grid dispatch, which inspires us to pay attention to the relationship between the prediction and subsequent decision-making cost. (Wang & Wu, 2017) discovered the asymmetry between cost and forecasting error and this asymmetry comes from actual scenarios. Underestimating predictions at peak points will result in additional power purchase costs while overestimating predictions at low points will waste power generation. Therefore, bias will be introduced if we just use traditional gradient loss functions like MSE and MAE to train the model. Then, (Zhang et al., 2022) proposed to use the characteristics of piecewise linearization and the Huber function to model the relationship between forecasting error and real cost. Inspired by this work, our package provides methods for modeling the relationship between forecasting error and other variables and then constructing the corresponding loss function.

To provide an accessible and extensible reference for future researchers in load forecasting, our developed package differs from the existing time series packages (Alexandrov et al., 2020; Godahewa et al., 2021). Specifically, our package splits the entire power load forecasting process into five modules: data preprocessing, feature engineering, forecasting methods, postprocessing, and evaluation metrics. Our package will cover both probabilistic forecasting and point forecasting, providing feature engineering methods and predictors based on traditional machine learning models and deep learning models. Users can combine any of these components and obtain their customized models. Furthermore, our package adds specific functionalities to address the characteristics of load forecasting and its differences from traditional time series forecasting, greatly enhancing the user's freedom to construct load forecasting models.

Lastly, we conduct extensive experiments to evaluate both point forecasting and probabilistic forecasting performance of different models on multiple load datasets at different levels. These results not only provide insights about different forecasting models under various scenarios and evaluation metrics but also show the accessibility and extensibility of our package and benchmark. Furthermore, we also demonstrate how the feature engineering and the loss function we provide could affect the load forecasting results.

**We summarize our primary contributions as follows**:

1. **Domain-specific feature engineering and self-defined loss function**. Based on the characteristics of load, temperature, and calendar variables, we integrate the feature engineering that reflects the ternary relationship into our package for users to use in any forecasting model. At the same time, we also provide users with a function to customize the loss function. Users can define the relationship between the forecasting error and any variable (such as the dispatching cost of the power grid) and integrate it into our forecasting framework as a loss function. In our experiment, we simulate an IEEE 30-bus system, provide the relationship between simulated forecasting error and cost, and construct the corresponding loss function.
2. **Fully open-source platform with accessibility and extensibility**. We release the relevant code on GitHub[1]. Users can freely combine the components we provide to design their load forecasting framework to cope with different power load forecasting scenarios. We provide over 20 forecasting methods, including both probabilistic forecasting and point forecasting. In addition, we also encourage users to combine their forecasting methods based on our framework. After defining the input and output of their method, users only need one command to add their forecasting model to our framework to accomplish common 24-hour-ahead electricity forecasting tasks. At the same time, we also provide various evaluation and visualization methods to facilitate users to evaluate the predictive performance of different models from multiple perspectives.

---

[1] https://anonymous.4open.science/r/ProEnFo-17CC

3. **The first large-scale benchmark for electrical load forecasting**. Based on **11** electrical load dataset including aggregated-level and building-level, we conduct our experiment on both probabilistic forecasting and point forecasting. For probabilistic forecasting, we compare **16** probabilistic forecasting methods and apply **11** metrics to comprehensively compare the performance. For point forecasting, we focus on **7** widely used deep learning methods and compare the traditional MSE loss function with our proposed loss function based on the relationship between forecasting error and cost. To the best of our knowledge, this is the first work to construct comprehensive benchmarks with large-scale datasets for load forecasting scenarios.

## 2   DATA DESCRIPTION

In this section, we will introduce how our dataset is collected and the characteristics of the dataset. We have collected a total of **11** datasets for our data collection, and a detailed description of each dataset is provided in the Appendix A. In summary, the data we collect mainly comes from UCI machine learning databases (Dua & Graff, 2017), Kaggle data competition platforms (Nicholas, 2019; Yeafi, 2021; Shahane, 2021), and the famous global energy forecasting competitions (Hong et al., 2014; 2016; 2019). In addition, we also include a dataset reflecting the impact of the COVID-19 epidemic on the power system into our archives. Under the influence of COVID-19, an influential external factor, the power load has changed significantly, posing a challenge to the robustness of the forecasting model (Farrokhabadi et al., 2022). From the perspective of load hierarchy, 7 of the data we collect are aggregated-level datasets, and the remaining 4 are building-level datasets. Aggregated-level load refers to the total load that aggregates multiple independent loads (such as the power demand of various buildings in the power system) together. More specifically, we classify the load of an area greater than one building as aggregated level. Because the aggregated-level load results from multiple load aggregations, it typically exhibits more pronounced periodicity and seasonality. For this reason, calendar variables significantly impact load forecasting at this level. In contrast, the load of the building level, which can also be seen as a part of the aggregated load. Building-level loads change very dramatically, resulting in significant uncertainty. Therefore, many works related to building-level load forecasting often focus on probabilistic forecasting (Xu et al., 2019; Jeong et al., 2021b). To provide a reference for researchers in related fields, we also collect building-level datasets from the Building Data Genome 2 (BDG2) Data-Set (Miller et al., 2020). In addition to different levels, the data we collect also has the characteristic of almost covering all meteorological data (actual measurement) such as temperature, which may be greatly beneficial to forecasting because of the great impact of external variables (especially temperature) on load. The number of time series contained in each dataset and their corresponding features are listed in Table 1. And all the data will be released under appropriate licenses.

Table 1: Datasets in the load forecasting archive.

| | Dataset | No. of series | Length | Resolution | Missing | License | Type | External variables |
|---|---|---|---|---|---|---|---|---|
| 1 | Covid19(Farrokhabadi et al., 2022) | 1 | 31912 | hourly | No | CC BY 4.0 | aggregated-level | airTemperature, Humidity, etc |
| 2 | GEF12(Hong et al., 2014) | 20 | 39414 | hourly | No | CC BY 4.0 | aggregated-level | airTemperature |
| 3 | GEF14(Hong et al., 2016) | 1 | 17520 | hourly | No | CC BY 4.0 | aggregated-level | airTemperature |
| 4 | GEF17(Hong et al., 2019) | 8 | 17544 | hourly | No | CC BY 4.0 | aggregated-level | airTemperature |
| 5 | PDB(Yeafi, 2021) | 1 | 17520 | hourly | No | CC0 1.0 | aggregated-level | airTemperature |
| 6 | Spain(Nicholas, 2019) | 1 | 35064 | hourly | Yes | CC0 1.0 | aggregated-level | airTemperature, seaLvlPressure, etc |
| 7 | Hog(Miller et al., 2020) | 24 | 17544 | hourly | Yes | MIT License | building-level | airTemperature, wind speed, etc |
| 8 | Bull(Miller et al., 2020) | 41 | 17544 | hourly | Yes | MIT License | building-level | airTemperature, wind speed, etc |
| 9 | Cockatoo(Miller et al., 2020) | 1 | 17544 | hourly | Yes | MIT License | building-level | airTemperature, wind speed, etc |
| 10 | ELF(Shahane, 2021) | 1 | 21792 | hourly | No | CC BY 4.0 | aggregated-level | No |
| 11 | UCI($Dua\&Graff$, 2017) | 321 | 26304 | hourly | No | CC BY 4.0 | building-level | No |

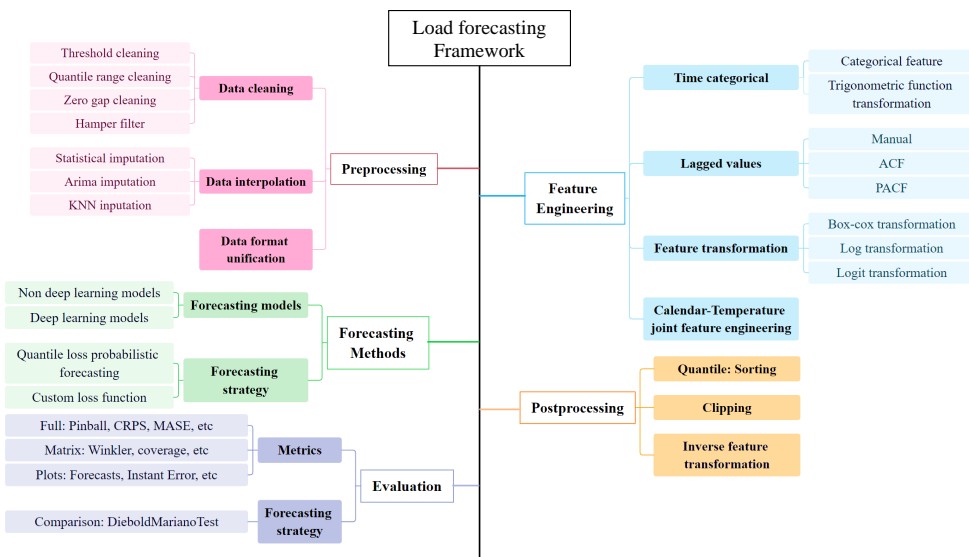

Figure 1: Overview of the load forecasting package.

# 3 PACKAGE FUNCTIONS

## 3.1 OVERVIEW OF THE PACKAGE

Fig 1 shows the overview of our package. As stated before, we divide the overall forecasting process into several parts to address potential issues in load forecasting for the power industry. First of all, load data is obtained by physical devices such as electricity meters. During this process, it is inevitable to encounter missing values, omissions, and other situations. Such a situation is more common in the load data of building-level (Jeong et al., 2021a). In this regard, our package provides various methods such as ARIMA based on Kalman filtering (Harvey & Pierse, 1984), K-nearest neighbor algorithm (García-Laencina et al., 2010) to fill in missing values, ensuring minimum data information distortion. Secondly, our model provides a variety of feature selection strategies to meet the needs of different scenarios. For example, users can choose the corresponding data from the previous seven days for day-ahead forecasting or use Autocorrelation Function (ACF) and Partial Autocorrelation Function (PACF) metrics to help select the lagged values. In addition, our framework allows users to add external variables such as temperature and calendar variables that may impact the forecasting model. As for the forecasting methods, we provide both probabilistic forecasting and point forecasting methods. Among them, probabilistic forecasting will be based on quantile forecasting. However, quantile regression may lead to confusion about quantile, that is, the forecasting result of a larger quantile is smaller than that of a smaller quantile. To address this situation, we have provided corresponding post-processing for reordering.

After obtaining forecasting results, we need reasonable metrics to evaluate them. Existing forecasting packages generally provide a variety of metrics, such as Pinball Loss and CRPS. Although they can evaluate the quality of forecasting results, they reduce the discrimination of forecasting models. For example, a model may perform poorly in a certain quantile while performing well in other quantiles. To more intuitively compare the performance of models in different quantiles, our package provides the matrix visualization function of multiple metrics in different quantiles. And the evaluation metrics we have implemented include CalibrationError (Chung et al., 2021), WinklerScore (Barnett, 1973), CoverageError, and so on (details can be seen in Appendix Section D.2).

## 3.2 TEMPERATURE-CALENDAR FEATURE ENGINEERING STRATEGY

The impact of temperature on load is greatly influenced by calendar variables. Inspired by the Hongtao linear regression model (Hong et al., 2016), we apply one-hot encoding to calendar variables and then model this coupling relationship by taking their products with temperature to the first,

second, and third powers as features. The specific formula is as follows.

$$\hat{y}_t = \beta_0 + \beta_1 \underbrace{\text{Trend}_t}_{\text{replaced by sequence model}} + \beta_2 M_t + \beta_3 W_t + \beta_4 H_t + \beta_5 W_t H_t + \beta_6 T_t + \beta_7 T_t^2$$

$$+ \beta_8 T_t^3 + \beta_9 T_t M_t + \beta_{10} T_t^2 M_t + \beta_{11} T_t^3 M_t + \beta_{12} T_t H_t + \beta_{13} T_t^2 H_t + \beta_{14} T_t^3 H_t,$$

where $M_t$, $W_t$, $H_t$, and $T_t$ represent the month, workday, and hour vectors after one-hot encoding and temperature at the corresponding time. Due to the one-hot coding, one categorical variable is changed into multiple binary categorical variables. When the corresponding variable is 0, the parameters of the linear model will not have any effect on it. Therefore, the result of doing so is constructing multiple personalized models based on calendar variables. Such a feature engineering strategy can help the forecasting model cope with situations where the temperature and load relationship shifts under different calendar variables. To preserve such characteristics and integrate existing sequence modeling methods (such as LSTM, and N-BEATS), we treat the information extracted by sequence modeling methods as trend variables and concatenate them with the previously obtained calendar temperature coupling variables. Finally, a fully connected layer is used to map the final output result. In section 5, we will compare the impact of this feature engineering on forecasting results across multiple datasets.

### 3.3 CUSTOM LOSS FUNCTION

Based on (Zhang et al., 2022), our framework provides corresponding piecewise linearization functions to help users model the relationship between forecasting errors and real requirements (such as scheduling costs) and integrate it into the gradient descent training. Specifically, we need data pairs $(\epsilon_i, C_i)_{i=1,...,N}$, where $\epsilon_i$ is the forecasting error and $C_i$ is the real requirement(here we mainly refer to the dispatching cost). Here, we consider using Forecasting Error Percentage (FEP) $\epsilon_i = \frac{f(x_i) - y_i}{y_i}$ as our error metric. At the same time, we normalize $\{C\}_{i=1,...,N}$, making its value fall between 0 and 1. Now, our goal has become how to construct $L(\epsilon)$ to estimate $C$. To achieve high fitting accuracy, we can use a spline cubic function, denoted as $s$, to fit it. However, the disadvantage of doing so is that there will be many discontinuities, which is not convenient to integrate them into our forecasting framework as a loss function. To ensure the fitting ability of the function while making it as simple as possible, a piecewise linearization strategy is adopted here. The number of segments K can be

determined by setting the upper bound of the fitting error $\|s - L(\epsilon)\|_2 \leq \frac{\left(\int_{\epsilon_{\min}}^{\epsilon_{\max}} s''(\epsilon)^{\frac{2}{5}} d\epsilon\right)^{\frac{5}{2}}}{\sqrt{120 K^2}}$ (Berjón et al., 2015). Regarding the position of the corresponding interval points, we strive to distribute the data points within the interval formed by each pair of endpoints (De Boor & De Boor, 1978). So far, we have obtained a piecewise linearization function. To take it as a loss function, we need to ensure its differentiability. Specifically, we use a quadratic function in a cell around each breakpoint to smooth it. Note that the quadratic function does not need to fit the data, but only needs to ensure its left and right continuity and the continuity of the corresponding first derivative to obtain the parameters.

Adapted from (Zhang et al., 2022), we employ a modified IEEE 30-bus test system to simulate an economic dispatch optimization problem (Hota & Naik, 2016). This system comprises 6 generators and 3 Battery Energy Storage Systems (BESS) connected to the network to supply power to 21 loads. We will address two dispatch optimization problems: the Day-Ahead Economic Dispatch (DAED) problem and the Intra-day Power Balancing (IPB) problem, to obtain the dispatch cost. The DAED problem primarily focuses on dispatching the power output of all online generators based on forecasted results, while the IPB problem fine-tunes the outputs of generators, charging/discharging behavior of BESS, and load shedding to balance the intra-day load deviation. The simulation process can be found

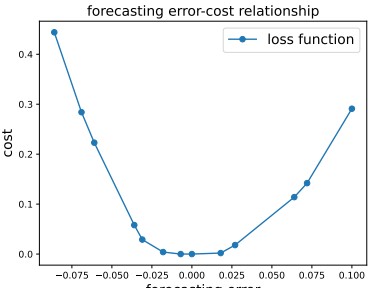

Figure 2: Visualization of simulated loss function.

in Appendix C.1. The simulated visualization result can be found in Fig 2. From the simulation results, we can identify two key distinctions in comparison to traditional Mean Squared Error (MSE) losses. First, there is a significant asymmetry between forecasting error and dispatching cost, which is evidenced by the considerable discrepancy in dispatching costs between instances when predicted

values are higher than the true value and instances when predicted values are lower than the true value. This difference stems from varying sources of dispatching costs. When the predicted value is too high, the system may over-dispatch power generation equipment, whereas when the predicted value is too low, the system may need to purchase emergency power supplies. Second, in order to minimize dispatching costs, we often do not need to require that predicted values perfectly match true values. Instead, we only need to meet specific accuracy requirements to minimize dispatching costs. This finding can guide the model to focus more on data points with low forecasting accuracy, thereby minimizing the final dispatching cost. We have packaged the loss function based on these simulated data for users to call.

### 3.4 Usage of the package

Our software package provides multiple components for users to freely combine and construct various power load forecasting scenarios. Meanwhile, we also encapsulate the default strategy, where users only need one command to achieve the default 24-hour ahead power probabilistic forecasting task and obtain detailed result files, including multiple forecasting methods and evaluation metrics for further analysis.

```
err_tot, forecast_tot, true = calculate_scenario(data=data,
target=target, methods_to_train=methods_to_train)
```

In addition, our package has a good extensibility and we also encourage users to implement more forecasting methods based on our framework and construct more power load forecasting scenarios. Similarly, after adding a new model, users can also add the new forecasting model to the forecasting task with a simple command.

```
methods_to_train.append(mi.MY_model())
```

We will provide a detailed description process in the Appendix C.2 and corresponding code documents to help users use our package to incorporate more power forecasting methods into the framework and construct different power load forecasting scenarios to adapt to different power forecasting tasks.

## 4 Benchmarking Process

In our archive, we will mainly discuss the results of probabilistic forecasting. At the same time, to explain our proposed custom loss function, we will also compare the point forecasting performance of the forecasting model trained using gradient descent.

**Data preprocessing and 24 hours-ahead forecasting**. We first use the functions provided by our framework to fill in missing values and address the issue of zero padding. For forecasting scenarios, we chose the most common load forecasting, which is to forecast 24 hours in advance, as our main task for evaluation (our framework also supports the construction of other forecasting scenarios). To meet the needs of subsequent power grid scheduling, load forecasting needs to reserve sufficient time for subsequent tasks, which means that there is a certain gap between the available historical sequences and the forecasting range. Therefore, we adopt the widely used forecasting setting in the power industry (Qin et al., 2023; Wang et al., 2022), which uses the historical values of the previous 7 days at the same time to forecast the corresponding power load on the 8th day.

**Feature engineering**. As mentioned in Section 3.2, we apply the transformation of feature engineering based on temperature and calendar variables to our forecasting models. For sequence models like the LSTM, we concatenate the features with the output of the models and input them into a single-layer MLP. As for the non-sequence models, we concatenate all the features and input lagged values. As a comparison, we also conduct experiments on non-transformed features simultaneously, directly inputting calendar variables and temperature as features.

**Forecasting models and loss functions**. We introduce 16 probabilistic forecasting methods for comparison, covering multiple types. These include two simple moving quantile methods based on global historical data (BEQ) and fixed-length windows (BMQ), as well as two models that account for forecasting errors, based on the Persistence (BECP) and linear regression methods (CE). In

addition, there are 5 non-deep learning methods, and they are quantile regression methods based on the K-nearest neighbor algorithm (Hastie et al., 2009), quantile regression methods based on random forest and sample random forest (Meinshausen & Ridgeway, 2006), and quantile regression methods based on extreme random tree and sample extreme random tree (Geurts et al., 2006). Finally, we introduce 7 deep learning methods, including simple forward propagation networks (Jain et al., 1996), LSTM networks (Hochreiter & Schmidhuber, 1997) for sequence modeling, convolutional neural networks (Li et al., 2021) (where we use one-dimensional convolutional kernels), Transformer (Vaswani et al., 2017) networks applying attention mechanisms. In addition, we include methods that modify the above neural network structures to make them more suitable for time series forecasting, such as LSTNet (Lai et al., 2018), which is designed to simultaneously capture both long-term and short-term patterns of time series, WaveNet based on causal convolution (Oord et al., 2016), and N-BEATS stacked into blocks using multiple linear layers (Oreshkin et al., 2020). Among them, the neural network is trained based on gradient descent. For probabilistic forecasting, we take the sum of ninety-nine quantile losses from 0.01 to 0.99 as the loss function. For point forecasting, we provide an asymmetric differentiable loss function through data fitting and integrate it into our forecasting framework as a loss function. At the same time, we also construct neural networks based on the traditional MSE Loss function for comparison.

## 5 BENCHMARK EVALUATION

With the help of the framework, we conduct extensive experiments on the collected load dataset based on the 16 probabilistic forecasting methods mentioned above. In addition, we also provide relevant point forecasting results for our proposed custom loss function. All experiments were conducted on Intel (R) Xeon (R) W-3335 CPU @ 3.40GHz and NVIDIA GeForce RTX3080Ti. Here, we primarily discuss the forecasting results of datasets with corresponding temperature data and the methods that can combine external data like temperature. The complete forecasting results as well as the running time of all the datasets are summarized in the Appendix D.3 and our code repository.

### 5.1 COMPARISON OF TEMPERATURE FEATURE ENGINEERING

In this section, we will examine the impact of the temperature feature engineering we provide on the forecasting results from the perspectives of both probabilistic forecasting and point forecasting. We use Pinball Loss↓ to evaluate the results of probabilistic forecasting and MAPE↓ to evaluate point forecasting.

Fig 3 reports partial probabilistic and point forecasting results, where the blue part represents the results of incorporating our feature engineering, while the green one represents the results without doing so. From the perspective of forecasting models, non-deep learning methods perform better than deep learning methods without the temperature transformation strategy. In deep learning methods, simple FFNN, LSTM, and CNN methods usually perform better than the more complicated ones. Moreover, complex deep learning models like the Wavenet and N-BEATS may even yield poor results. With the temperature transformation strategy, non-deep learning methods do not experience much improvement. The KNNR method experienced significant performance degradation on all datasets. This is because our feature engineering makes the input features very sparse, which seriously affects the performance of the K-nearest neighbor clustering algorithm, leading to a decrease in performance. However, deep learning methods have great improvements with this feature engineering.

Table 2 summarizes partial probabilistic forecasting results with PinBall Loss↓ under different deep models. Among them, for the COVID-19 dataset, adding feature engineering significantly worsens the result. The characteristic of this data is that after the impact of COVID-19, the load of the power system has changed significantly, and there is a large deviation between the training set and the test set. Therefore, the decrease in forecasting performance indicates that after this feature engineering, the model tends to learn more about the relationship between temperature and load, while ignoring the influence of historical load to a certain extent. In addition, the probabilistic forecasting results in the Spanish dataset also indicate the negative effect of the feature engineering. This is because the temperature data of this dataset does not exactly match the load data (see Appendix A for details). On the contrary, in datasets such as GEF12, 14, and 17, it can be seen that for relatively stable aggregated level loads with corresponding temperature data, such feature engineering can significantly improve the performance of the forecasting model.

We provide more complete results in the Appendix D.3 and the corresponding code repository.

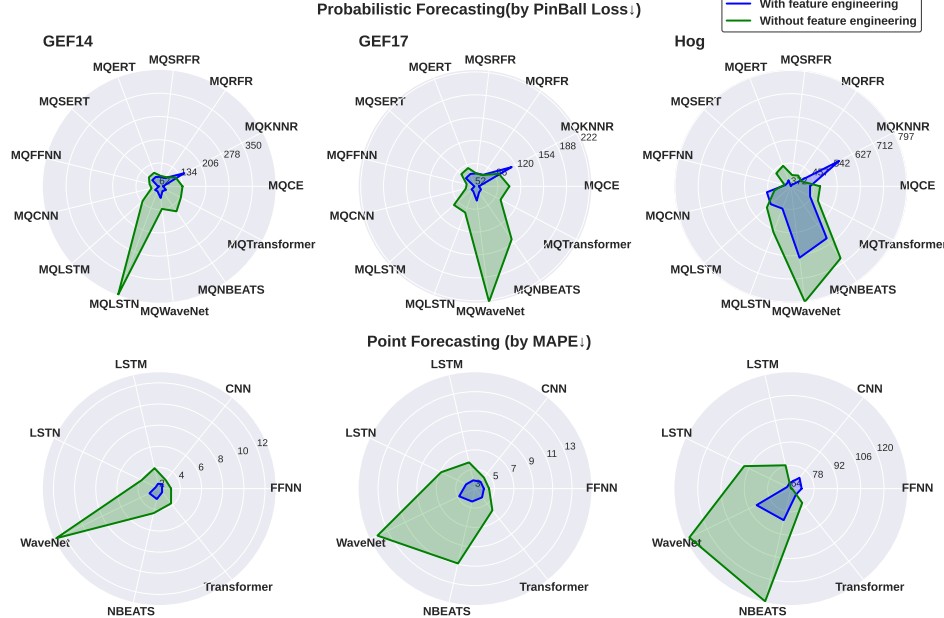

Figure 3: Comparison on parts of datasets (note that GEF14 and GEF17 are aggregated-level while Hog is building-level).

Table 2: Comparison of temperature feature engineering on deep learning-based probabilistic forecasting by PinBall Loss (the lower the better, the underline indicates a performance improvement after incorporating temperature-calendar feature engineering).

(a) Without temperature feature engineering (partial)

| Pinball Loss ↓ | FFNN | LSTM | CNN | Transformer | LSTNet | N-BEATS | WaveNet |
|---|---|---|---|---|---|---|---|
| Covid19 | 21,964.79 | 20,491.07 | 18,785.34 | 21,536.55 | 21,532.66 | 25,760.11 | 24,242.31 |
| GEF12 | 6,197.50 | 8,345.04 | 6,147.87 | 8,536.78 | 8,492.35 | 8,806.05 | 15,589.50 |
| GEF14 | 92.99 | 131.05 | 86.51 | 137.47 | 421.25 | 156.07 | 132.32 |
| GEF17 | 64.81 | 93.90 | 66.26 | 94.83 | 93.58 | 147.78 | 224.84 |
| Spain | 1,503.36 | 1,656.65 | 1,389.27 | 1,764.79 | 1,666.29 | 1,458.66 | 3,402.79 |
| Bull | 12.80 | 13.90 | 12.20 | 14.16 | 15.31 | 19.87 | 21.71 |
| Hog | 396.26 | 489.46 | 433.83 | 483.95 | 552.26 | 693.08 | 798.83 |
| Cockatoo | 51.10 | 68.46 | 35.45 | 67.21 | 67.70 | 122.90 | 122.97 |
| PDB | 404.23 | 654.04 | 369.75 | 619.95 | 645.56 | 1,902.73 | 1,933.24 |

(b) With temperature feature engineering (partial)

| Pinball Loss ↓ | FFNN | LSTM | CNN | Transformer | LSTNet | N-BEATS | WaveNet |
|---|---|---|---|---|---|---|---|
| Covid19 | 33,109.56 | 30,194.04 | 34,247.64 | 33,502.34 | 34,793.91 | 74,741.26 | 43,739.91 |
| GEF12 | 5,880.49 | 6,327.38 | 5,858.94 | 6,514.78 | 6,142.80 | 6,933.65 | 7,544.14 |
| GEF14 | 67.36 | 79.26 | 62.05 | 85.93 | 77.99 | 79.70 | 98.37 |
| GEF17 | 53.53 | 59.05 | 52.82 | 62.08 | 59.77 | 62.66 | 73.37 |
| Spain | 1,680.22 | 1,784.30 | 1,544.18 | 1,703.50 | 1,623.11 | 2,053.57 | 1,620.20 |
| Bull | 12.23 | 12.89 | 12.55 | 13.76 | 13.60 | 15.25 | 15.34 |
| Hog | 392.52 | 470.77 | 462.54 | 451.55 | 459.44 | 603.73 | 635.95 |
| Cockatoo | 35.66 | 43.34 | 34.11 | 42.85 | 42.38 | 48.78 | 41.91 |
| PDB | 263.59 | 361.91 | 255.95 | 387.89 | 347.62 | 359.84 | 358.55 |

## 5.2 COMPARISON OF ASYMMETRIC FITTING LOSS FUNCTION

According to (Zhang et al., 2022), the relationship between load error and the scheduling cost it causes is not symmetrical; the cost of underestimating at peak and overestimating at low values is

Table 3: Comparison of different loss functions on deep learning-based probabilistic forecasting by cost (see the description in section 3.3).

(a) Trained by MSE loss function

| Cost↓ | FFNN | LSTM | CNN | Transformer | LSTNet | N-BEATS | Wavenet |
|---|---|---|---|---|---|---|---|
| Covid19 | 0.2048 | 0.1965 | 0.2242 | **0.1749** | 0.2911 | 0.6141 | 0.6140 |
| GEF12 | 0.3259 | 0.3213 | 0.3337 | 0.3298 | 0.3314 | 0.3571 | 0.4438 |
| GEF14 | 0.0384 | 0.0417 | 0.0413 | 0.0421 | 0.0412 | 0.0648 | 0.0651 |
| GEF17 | 0.1109 | **0.0982** | 0.1028 | 0.1095 | 0.1075 | 0.1177 | 0.1614 |
| Spain | 0.2248 | 0.2192 | 0.2244 | **0.2025** | 0.2067 | 0.2038 | 0.2684 |
| Bull | 1.8616 | 1.7499 | 1.8071 | 1.7603 | 1.7768 | 1.7765 | 2.2614 |
| Hog | 1.4099 | 1.3431 | 1.2443 | **1.1334** | 1.3918 | 1.5175 | 1.8091 |
| Cockatoo | 1.7939 | 1.4710 | 1.8784 | 1.4991 | 1.3170 | 1.4124 | 1.7414 |
| PDB | 0.2487 | 0.1808 | 0.1848 | 0.1733 | 0.1906 | 0.1568 | 0.2412 |

(b) Trained by asymmetric fitting loss function

| Cost↓ | FFNN | LSTM | CNN | Transformer | LSTNet | N-BEATS | Wavenet |
|---|---|---|---|---|---|---|---|
| Covid19 | 0.1977 | 0.1866 | 0.2005 | 0.2238 | 0.2308 | 0.2242 | 0.6949 |
| GEF12 | 0.3227 | 0.3324 | **0.3207** | 0.3412 | 0.3365 | 0.3542 | 0.4178 |
| GEF14 | **0.0380** | 0.0461 | 0.0392 | 0.0422 | 0.0703 | 0.0715 | 0.0707 |
| GEF17 | 0.1352 | 0.1165 | 0.1298 | 0.1272 | 0.1287 | 0.1792 | 0.1728 |
| Spain | 0.2301 | 0.2340 | 0.2276 | 0.2441 | 0.2142 | 0.2318 | 0.2163 |
| Bull | 1.8245 | 1.7592 | 1.7679 | **1.7314** | 1.8759 | 1.7930 | 2.1777 |
| Hog | 1.3157 | 1.2560 | 1.4364 | 1.2189 | 1.4511 | 1.3205 | 1.5243 |
| Cockatoo | 1.2561 | **1.1589** | 1.1991 | 1.2367 | 1.2486 | 1.2493 | 1.2455 |
| PDB | **0.0449** | 0.0597 | 0.0451 | 0.0583 | 0.0608 | 0.1192 | 0.1211 |

different. Therefore, we use the relationship between forecasting error and cost $(\epsilon_i, C_i)$ as shown in Figure 2 to estimate the costs of different methods and evaluate the impact of our asymmetric loss function. Among them, because the results in the previous section show that temperature-based feature engineering significantly improves the deep learning network, we apply this feature engineering to all of the methods.

Table 3 reports the point forecasting results with different loss functions. It can be seen that asymmetric loss brings improvements in most cases. However, in the GEF17 and Spain datasets, the performance of the asymmetric loss function is generally lower than that of the MSE loss function. Due to our estimation of dispatching costs coming from a 30-bus system, there may be some differences in scale compared to a real large-scale power system, so our loss function may perform poorly on aggregated-level data like the GEF17. On the contrary, with building-level data, our custom loss function can help the model recognize the asymmetric nature between forecasting error and cost, bringing significant positive effects on minimizing costs. Overall, asymmetric loss functions can provide some performance improvement for most methods on most datasets, thereby minimizing the dispatching costs.

## 6 CONCLUSIONS

In this paper, we construct a package and benchmark specifically designed for electrical load forecasting. Our load forecasting package comes with high accessibility and extensibility by dividing the entire power forecasting process into several modules for users to freely combine and construct their own forecasting frameworks. In addition, our package also provides the engineering implementation of features based on temperature and the construction method of custom loss functions by data fitting. Meanwhile, with the help of our package, we have provided comprehensive forecasting benchmark results using multiple forecasting methods and multiple datasets as well as detailed discussion and analysis, serving as an important reference for researchers in the community.

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

# A  DATASET DESCRIPTION

As shown in Table 1, we collected multiple load datasets at different levels and organized them into a user-friendly '.pkl' format and users can obtain them through the URL at GitHub we provide. Now we will introduce their sources and detailed information one by one.

## A.1  GEF12

The GEF12 dataset is sourced from the Global Energy Forecasting Competition 2012 (Hong et al., 2014). This competition has multiple tracks, and we have compiled the dataset provided by the load forecasting tracks as one of our benchmark datasets. In this dataset, there are a total of 20 aggregated-level load series data and 11 temperature series. It is worth noting that the one-to-one correspondence between these temperature data and load data has not been clearly defined. For simplicity, the strategy used in our benchmark testing is to simply use only one temperature series data as the temperature variable for all series (the randomly selected result here is the second temperature data). Each time series covers load data with a resolution of 1 hour from 0:00 on January 1, 2004, to 5:00 on June 30, 2008. Because this dataset is used for competitions and the integrity of the data is relatively good, we did not preprocess the data (such as filling in missing values).

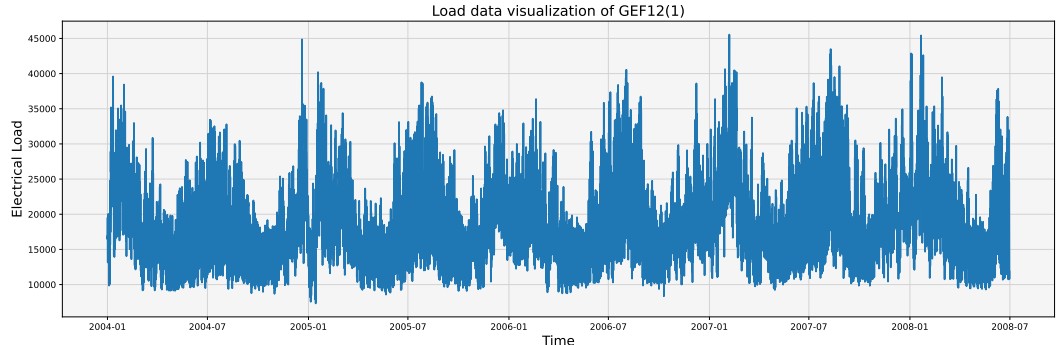

Figure 4: Visualization of the first load series in the GEF12 dataset.

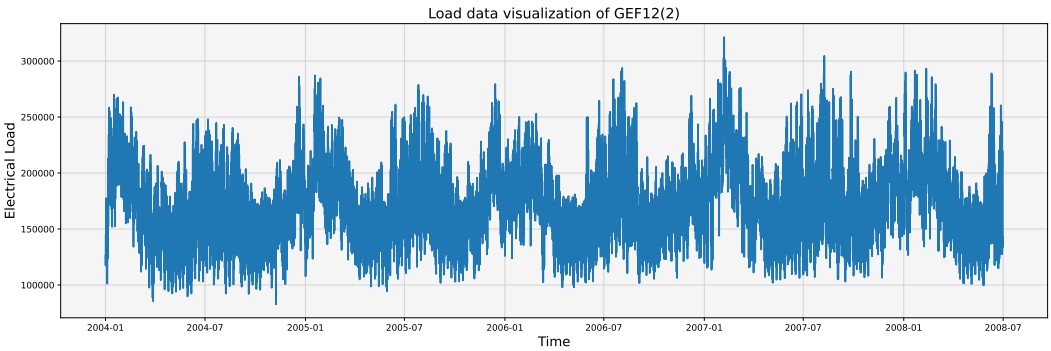

Figure 5: Visualization of the second load series in the GEF12 dataset.

Fig. 4 and Fig. 5 visualize the partial load sequence in the GEF12 dataset. From it, we can see that these load series have obvious periodicity and seasonality. And this is an important feature of aggregated-level load.

## A.2  GEF14

The GEF14 dataset is from the Global Energy Forecasting Competition 2014 (Hong et al., 2016). This competition also has multiple tracks, and we focus on load forecasting tracks. The competition provides load data spanning up to 8 years from 2006 to 2014. Unlike the 2012 competition, We

truncate the load data and only use the data from 2013 and 2014 for testing. On the one hand, it is because the impact of load data from many years ago on the current forecast is very small, and on the other hand, it is because most of the load data we collect is about 2 years in length. For relative consistency, we only took the last two years to construct our load forecast archive. Our adjusted load data covers load data with a resolution of 1 hour from 1:00 on January 1, 2013, to 0:00 on January 1, 2015. Fig. 6 shows the adjusted load data, similar to the data in GEF12, which is also aggregated level data. Therefore, the data in GEF14 also shows obvious periodicity and seasonality.

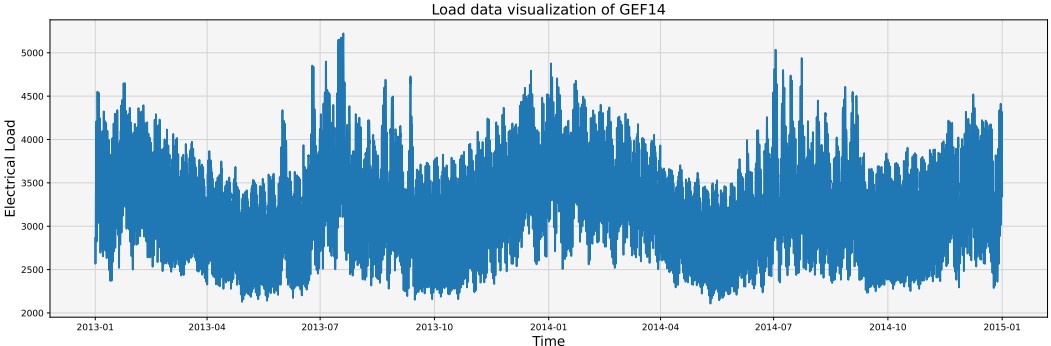

Figure 6: Visualization of the load series in the GEF14 dataset.

### A.3 GEF17

The GEF17 dataset is from the Global Energy Forecasting Competition 2017 (Hong et al., 2019). Similar to GEF12, this dataset also provides multiple aggregated-level load data. However, the difference is that it clarifies the corresponding relationship between the temperature series and load series, providing a one-to-one temperature series corresponding to the load series. In terms of period, it provides load data from 2013 to 2017. For the reasons mentioned above, we have intercepted the load data and only used the load data from the past two years (i.e. 2016 and 2017). Finally, we used 8 aggregated-level load data from 2016 to 2017 and their corresponding temperature data to construct our load forecasting archive. Fig. 7 and Fig. 8 visualize some data in the GEF17 dataset, similarly, it also showcases the common characteristics of aggregated-level load data, namely periodicity and seasonality.

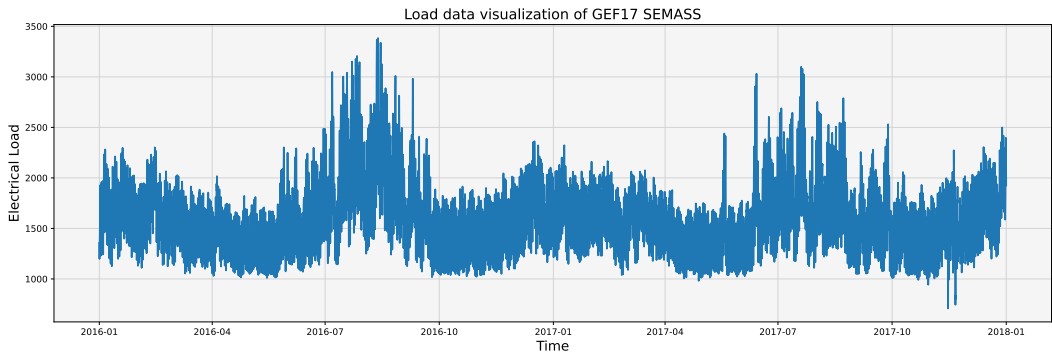

Figure 7: Visualization of the load series in the GEF17 SEMASS dataset.

### A.4 COVID19

The Covid19 dataset is from the Day-Ahead Electricity Demand Forecasting Competition: Post-COVID Paradigm (Farrokhabadi et al., 2022). This dataset covers the load data from 0:00 on March 18, 2017, to 15:00 on November 5, 2020. In addition to load data and temperature, this dataset also provides other meteorological factors such as humidity and wind speed. To maintain the consistency

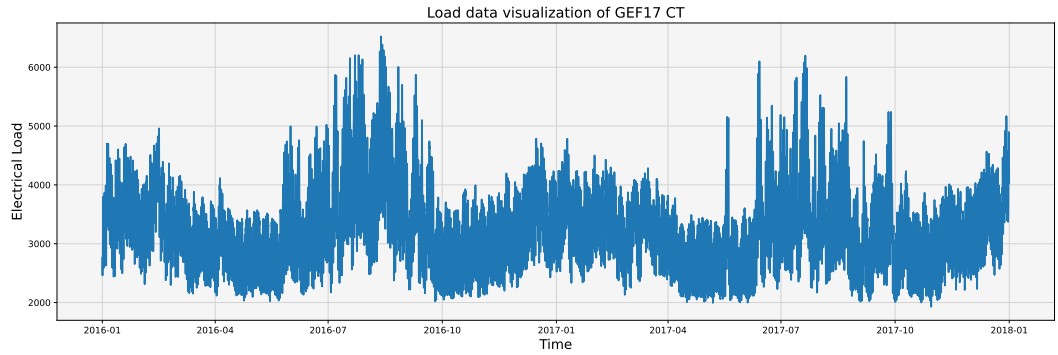

Figure 8: Visualization of the load series in the GEF17 CT dataset.

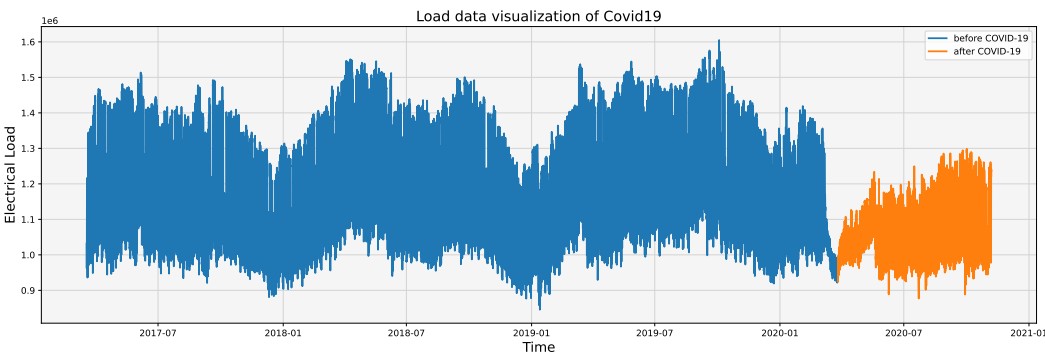

Figure 9: Visualization of the load series in the Covid19 dataset.

of the forecasting archives, we did not consider such factors. Unlike the datasets mentioned above, this dataset focuses on the impact of COVID-19 on the power system. Fig. 9 shows the load data in the Covid19 dataset. The blue part indicates that the power system has not yet been impacted by COVID-19, similar to other aggregated level load data, showing periodicity. The orange section displays the load data after COVID-19. It can be seen that it is different from the blue part. The absolute value of the load rapidly decreases during the period being impacted, and then recovers smoothly after a period of time. However, compared to the same period when it was not impacted, the load value has decreased. This transformation poses a challenge to the robustness of forecasting models. As shown in the main text, our temperature-calendar variable feature engineering will make the model more inclined to remember the impact of the temperature and calendar variable on the load and ignore the historical value to a certain extent, which ultimately leads to the decline of the forecasting performance. Therefore, when encountering strong external events like this, day-ahead forecasting should focus more on historical values.

## A.5 PDB

The PDB dataset is a public dataset from the Kaggle data competition platform (Yeafi, 2021). It covers load and temperature data from 1:00 on January 1, 2013, to 0:00 on January 1, 2015. Due to its moderate length, we did not intercept it. Fig. 10 shows its load data visualization results.

## A.6 SPANISH

The Spanish dataset is also a public dataset from the Kaggle data competition platform (Nicholas, 2019). It provides nationwide load data for Spain from 0:00 on January 1, 2015, to 23:00 on December 31, 2018. At the same time, it also provides meteorological data (such as temperature and wind speed) corresponding to the five major cities in Spain. This situation is similar to GEF12. For the same reason, we select relevant meteorological data from the economically developed Barcelona region as

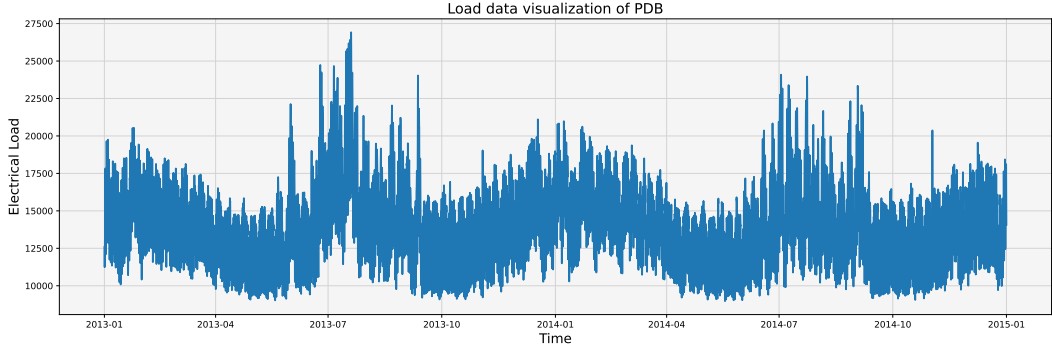

Figure 10: Visualization of the load series in the Spanish dataset.

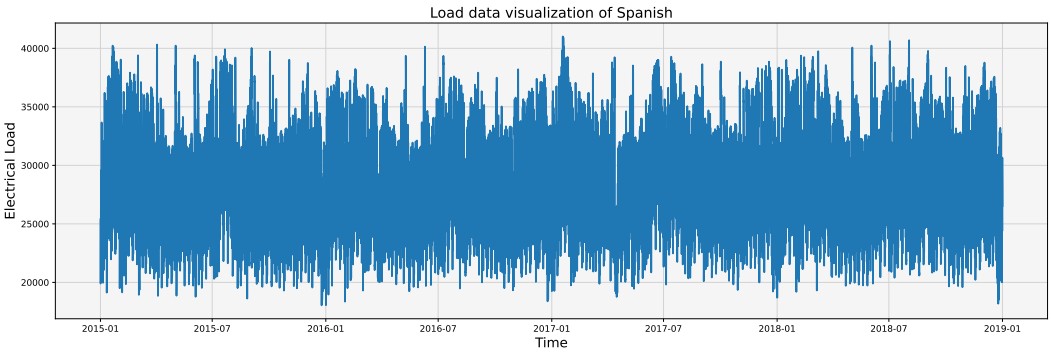

Figure 11: Visualization of the load series in the PDB dataset.

the corresponding meteorological data. In addition, the load data of this dataset is partially missing (with a missing rate of 0.1%). Because of the low missing rate, we used a simple Linear interpolation method to fill the data. Fig. 11 shows the corresponding load visualization results. Compared to other aggregated level loads, the periodicity and seasonality of the Spanish national load have become relatively less pronounced.

### A.7 HOG

The Hog dataset comes from The Building Data Genome 2 (BDG2) Data-Set (Miller et al., 2020). BDG2 is an open dataset that includes building-level data collected from 3053 electricity meters, which covers 1636 buildings. From the perspective of the area where the building is located, it includes the load, cooling, and heating data of buildings in multiple areas such as Hog and Bull. From a period perspective, it covers data from 2016 and 2017. In addition, BDG2 also classifies buildings, including buildings for educational purposes, offices, and so on. Based on the characteristics of this dataset, we divide it by region, and the Hog dataset is composed of relevant load data from buildings in the Hog region in the BDG2 dataset. Because the data in this dataset is all building-level data and we often find situations such as missing values and outliers in data at this level(Jeong et al., 2021a). Therefore, we first use the functions provided by the package to check for outliers. Specifically, we first calculate the lower quartile (Q1) and the upper quartile (Q3) and then calculate the quartile interval (IQR), that is, $IQR = Q3 - Q1$. Here, the outlier is defined as the point that is lower than $Q1 - q \times IQR$ or higher than $Q3 + q \times IQR$. The outlier factor $q$ here is set to 1.5. We set the detected outlier as the missing value, and discard the sequence with a missing rate of more than 10%. For sequences with a missing rate of less than 10%, we interpolate them (using linear, polynomial, etc.). Finally, we obtained 24 available load sequences and their corresponding temperature sequences for the Hog region.

Fig. 12 and Fig. 13 show two load sequences in the Hog dataset. They belong to educational facilities and offices respectively. It can be seen that compared to aggregate-level datasets, building-level

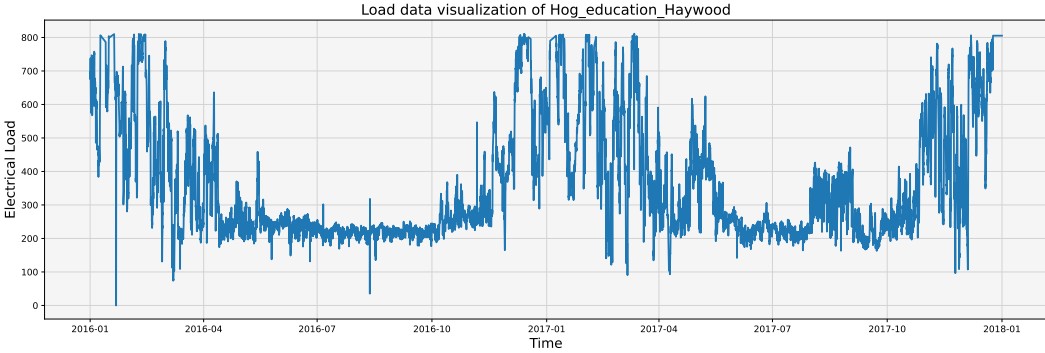

Figure 12: Visualization of the Hog education Haywood in the Hog dataset.

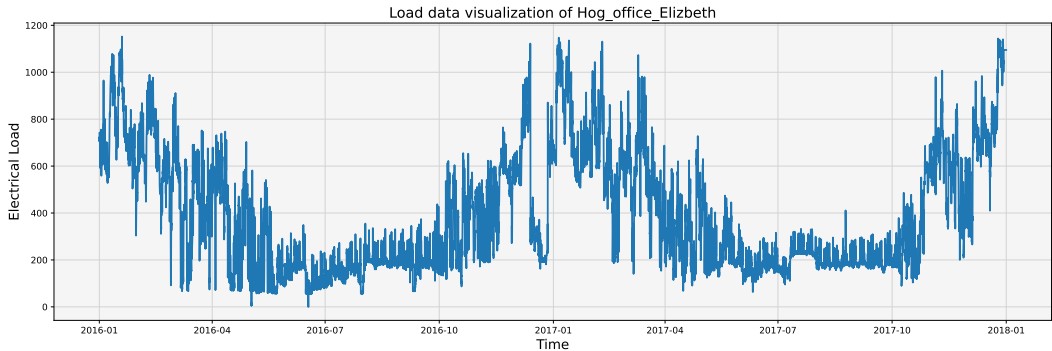

Figure 13: Visualization of the Hog office Elizbeth series in the Hog dataset.

datasets exhibit greater uncertainty. The similarity of data for the same period in different years is also significantly lower than those aggregate-level ones. Although the data is only two years old, the building dataset also exhibits significant seasonality. Specifically, the load during summer and autumn is relatively high, while the load during winter and spring is relatively low. In addition, despite the different properties of buildings, they still maintain a relatively similar seasonality.

### A.8  BULL

Similar to the Hog dataset, the Bull dataset also comes from the BDG2 dataset. Similarly, we screen and preprocess the building load data in the Bull area, resulting in 41 available sequences covering multiple building properties. Fig. 14 and Fig. 15 show the load data of two representative building types in the Bull area. Similar to other building-level load data, the manifestation of periodicity is not obvious. Meanwhile, sudden changes also occur from time to time, posing challenges for the forecasting model to accurately model and forecast.

### A.9  COCKATOO

Cockatoo is also from the BDG2 dataset. However, after our screening, only one load sequence met our requirements, which is "Cockatoo Office Laila". Fig. 16 shows the load characteristics of this building. It is worth noting that during the period from February to April 2016, the load data appeared relatively stable. This may be caused by the fault of the measuring meters, by human error in the reading, or it may be the real situation. These errors typically occur in building-level load data and it is difficult to avoid this situation through data cleansing unless we directly discard the relevant data. Here, we retain this data to evaluate its impact on the final forecasting performance.

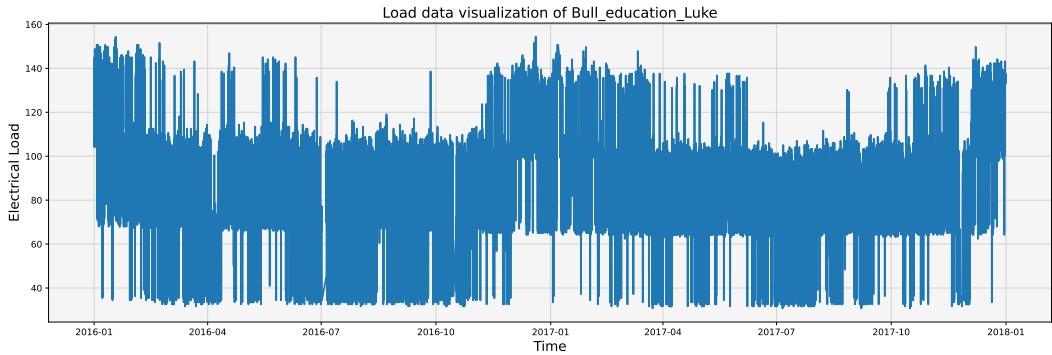

Figure 14: Visualization of the Bull education Luke series in the Bull dataset.

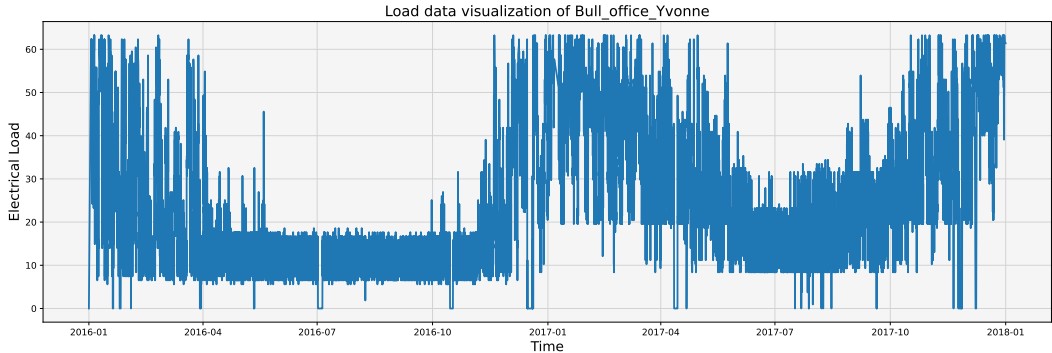

Figure 15: Visualization of the Bull office Yvonne series in the Bull dataset.

### A.10   ELF

The ELF dataset comes from the Kaggle data platform (Shahane, 2021). It is worth noting that the platform provides temperature data from multiple Panama cities as well as other meteorological data such as wind speed and humidity. However, the relationship between these meteorological data and load data has not been clarified, and unlike datasets such as Spanish, we are not clear about the detailed regions to which the load value data belongs. Therefore, we only conduct experiments on historical load series and calendar variables. However, users can also add relevant meteorological variables to the forecasting model through simple code. Fig. 17 shows the load data for this dataset. Similar to the Covid19 dataset, this dataset also shows the impact of COVID-19 on the power system (see the data after April 2020).

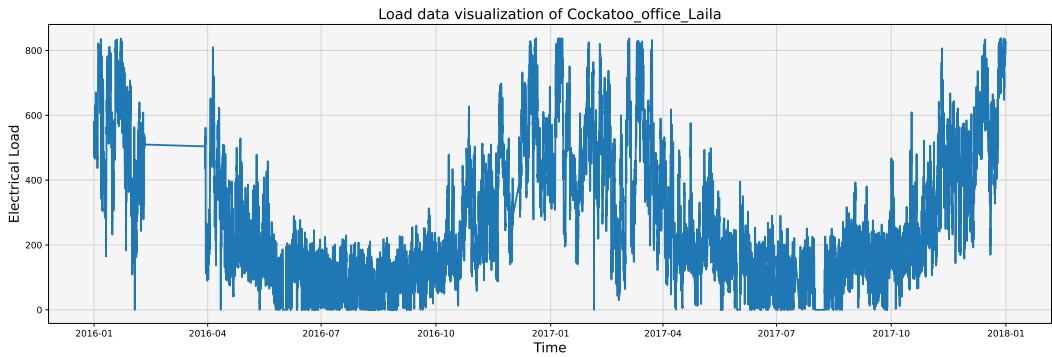

Figure 16: Visualization of the Cockatoo office Laila series in the Cockatoo dataset.

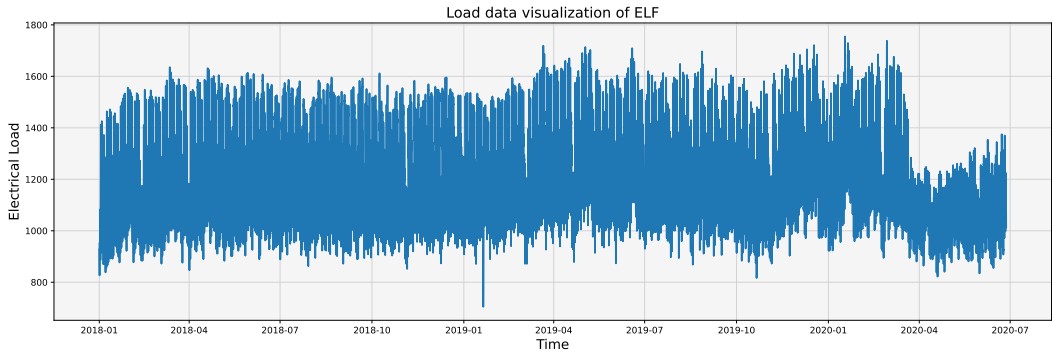

Figure 17: Visualization of the load series in the ELF dataset.

## A.11  UCI

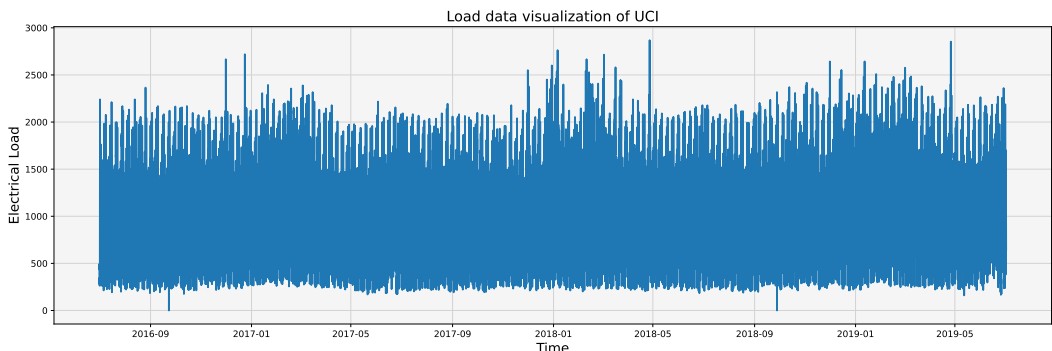

Figure 18: Visualization of the load series in the UCI dataset.

The UCI dataset is a power load dataset from the UCI database, which has been widely used in the field of machine learning(Dua & Graff, 2017). Unlike the original dataset, we select the processed version, which includes 321 load sequences with hourly resolution (Bergsma et al., 2022).

## B  FEATURE ANALYSIS

As we mentioned before, external features have a significant impact on load forecasting. And among them, temperature variables and calendar variables have the greatest impact. This is also recognized by the famous global energy forecasting competition. Based on this, the organizer developed a linear model called the HongTao vanilla model, which considers load, calendar variables, and temperature as the main variables and it serves as the benchmark for the forecasting competition (Hong et al., 2014). Therefore, in this section, we will visualize the relevant features in different levels of datasets to explore the relationship between load, calendar variables, and temperature. At the same time, we will also provide feature engineering based on the relationship among load, calendar variables, and temperature.

### B.1  TEMPERATURE-LOAD ANALYSIS

Figures 19 and 20 show scatter plots of the relationship between temperature load at two different levels, respectively. The data at the aggregated level comes from the GEF14 competition, while the building level is randomly selected from the BDG2(Bull) dataset. It is worth noting that in the BDG2 dataset, each building has its corresponding attribute usage, such as educational facilities, office space, and so on. We divided the scatter plot of load and temperature into 12 blocks by month, with the aim of expanding the relationship between temperature and load to the ternary relationship

between temperature, load, and calendar variables. Among them, we can consider calendar variables as indicators of seasons, months, workdays (weekends), and hours, and explore the temperature load relationships of different seasons (months, etc.). Similarly, we will only analyze the months here and include the remaining analysis.

From figure 19, we can see that, in line with common sense, the relationship between load and temperature shows significant differences when in different months. From May to September, there is a significant positive correlation between load and temperature. Starting from October, the relationship between load and temperature gradually shifted from a significant positive correlation to an insignificant correlation. May to September is also a period of frequent high-temperature weather, indicating that when the temperature is high, there is a significant positive correlation between temperature and load. It is worth noting that this positive correlation is not always true. If we directly hand over the temperature variables to the model for modeling without processing, such changes in the relationship may cause confusion and ultimately lead to a decrease in forecasting performance.

When it comes to building-level load, figure 20 shows that the uncertainty of the detected load is significantly greater than that of the aggregated load since each month presents different temperature-load relationships. As this is the load data from educational facilities, there may be classifications such as teaching days or rest days, as shown in the figure, where there is a clear phenomenon of fragmentation within multiple months.

In summary, the two levels of load exhibit different load-temperature relationships in different months. This situation also occurs in other time scales such as the hour. Therefore, to make the forecasting model understand this relationship correctly, it is necessary to consider calendar variables and temperature together.

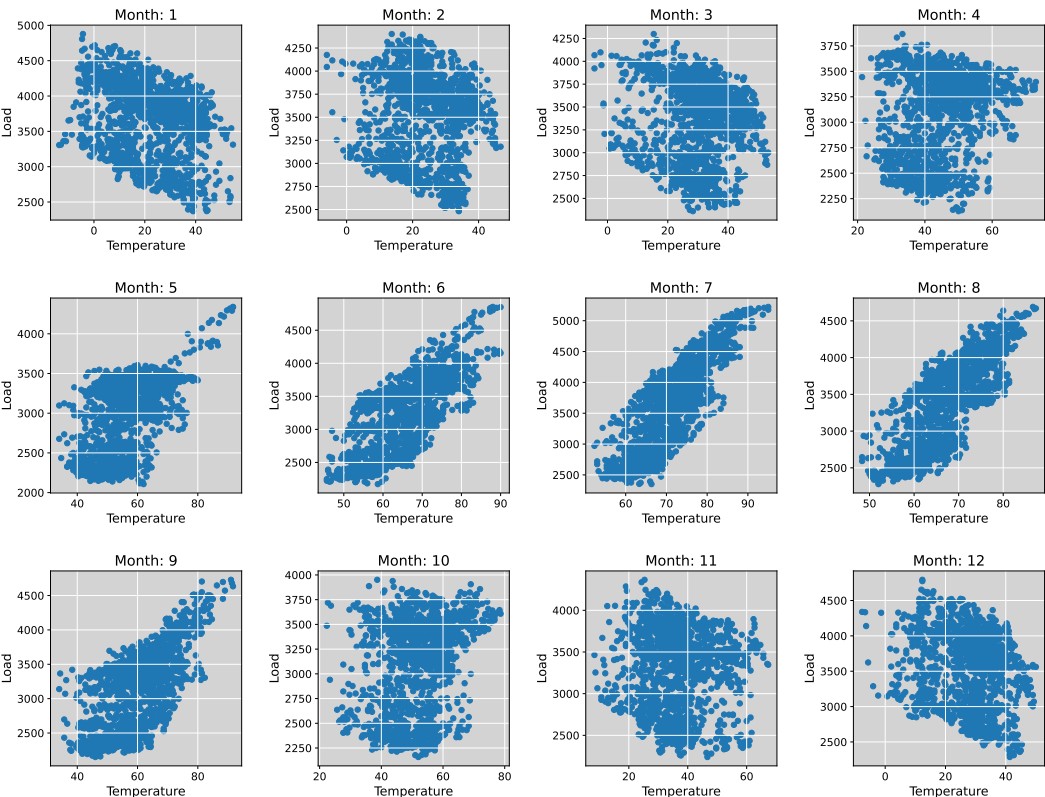

Figure 19: Temperature-aggregated load scatter plots for 12 months.

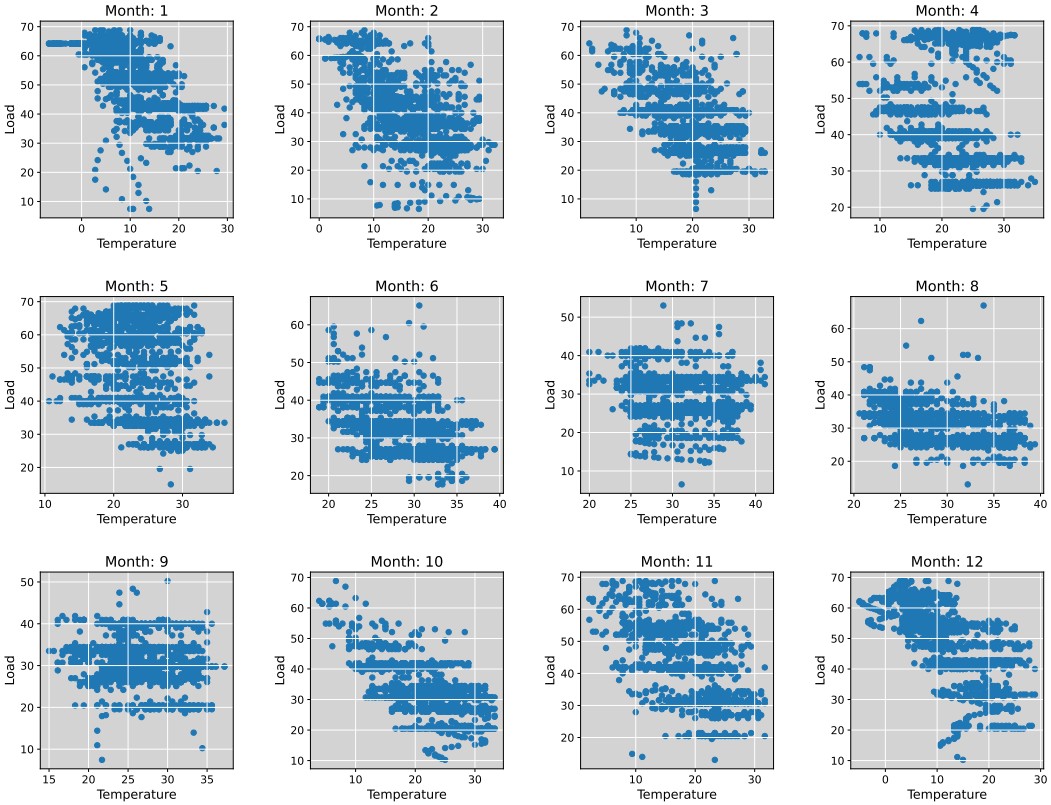

Figure 20: Temperature-building load scatter plots for 12 months.

## C PACKAGE USAGE

### C.1 ASYMMETRIC LOSS FUNCTION

The ultimate goal of power load forecasting is to minimize subsequent scheduling costs, which are closely related to prediction errors. Inspired by (Zhang et al., 2022), our package provides a piecewise linearized function to fit forecasting errors with other variables (which can be corresponding scheduling costs, etc.). At the same time, we also give an asymmetric loss function to replace the symmetric MSE loss function. Specifically, in actual power grid dispatch, the economic losses caused by forecasting values being less than the true values are often greater than the losses caused by forecasting values being greater than the true values. Therefore, we use a simple quadratic function to construct a piecewise generating function.

Here, $\epsilon$ represents the Forecasting Error Percentage(FEP) $\epsilon = \frac{f(x)-y}{y}$. We first use this function to sample and obtain many data points and then use a smoothing spline, denoted as $s$, to fit them. To avoid many breakpoints, which may make it difficult to integrate into our forecasting framework as a loss function (Perperoglou et al., 2019), we use piecewise linearization to approximate the spline function. The selection of breakpoints can be based on the following formula(Berjón et al., 2015),

$$\|s - L(\epsilon)\|_2 \leq \frac{\left(\int_{\epsilon_{\min}}^{\epsilon_{\max}} s''(\epsilon)^{\frac{2}{5}} d\epsilon\right)^{\frac{5}{2}}}{\sqrt{120K^2}},$$

where K is the number of breakpoints, and the integration interval we choose here is $(-0.15, 0.15)$. By controlling the error between piecewise linear functions and spline functions, we can obtain an appropriate number of breakpoints. Here we control the error lower than 0.005. As for the location of the breakpoints, we first calculate the cumulative breakpoint distribution function according

to (De Boor & De Boor, 1978),

$$F\left(\epsilon_k\right) = \frac{\int_{\epsilon_{\min}}^{\epsilon_k} |s''(x)|^{2/5}\, dx}{\int_{\epsilon_{\min}}^{\epsilon_{\min}} |s''(x)|^{2/5}\, dx}.$$

The breakpoints $\{\epsilon_k\}_{k=1}^{K-1}$ will be placed such that each subinterval can contribute equally to the value of the cumulative breakpoint distribution function. To eventually integrate it into our forecasting framework as a loss function, we need to ensure that it is differentiable. And we can achieve this by inserting a quadratic function at each breakpoint. Specifically, we insert a quadratic function within the 0.000001 distance before and after each breakpoint and obtain the parameters of the quadratic function by ensuring the continuity of the function and its first derivative at the two connections before and after.

As mentioned in section 3.3, we simulate an IEEE 30 bus system to get the relationship between forecasting error and dispatching cost. Fig 21 shows the specific process of simulation. Here we mainly focus on two optimization problems DEAD and IPB. The mathematical definitions of these two can be found in (Zhang et al., 2022). To solve these two optimization problems, we have provided the corresponding MATLAB code. Based on the above process, we have provided the relationship between the load forecasting error and the actual dispatching cost for each hour within 24 hours of a day, as shown in Fig 22. All corresponding data is saved in the file "breakpoint_new.mat" we provide and we can construct the corresponding loss function through a single line of code. Note that our experiment is conducted based on the hour 9.

```
loss_function = ContinuousPiecewiseLinearFunction(breakpoint)
```

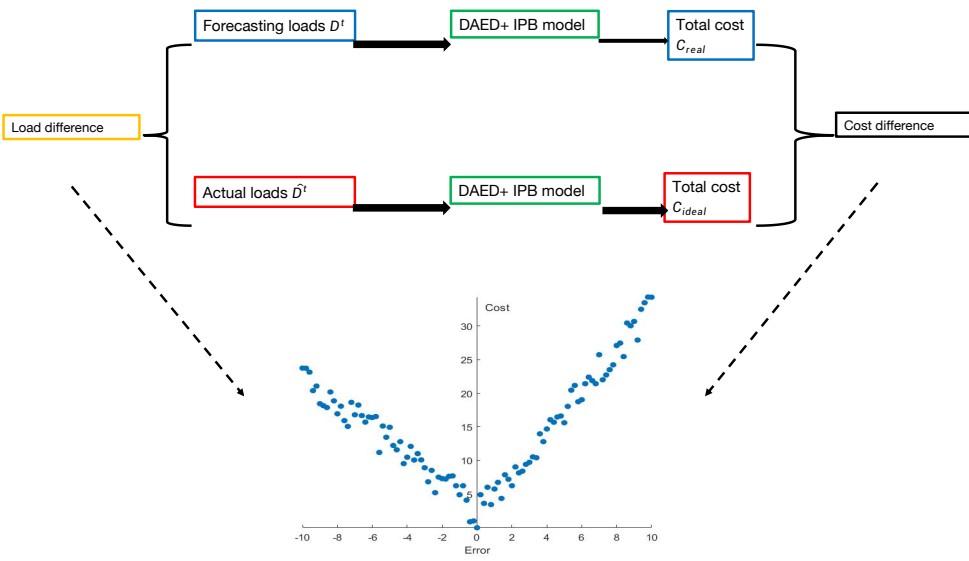

Figure 21: Loss function generation (adapted from (Zhang et al., 2022)).

## C.2 HOW TO USE THE PACKAGE

### C.2.1 HOW TO CONSTRUCT FORECASTING SCENARIOS

Our framework mainly constructs forecasting scenarios through the function "calculate_scenario". Below, we will introduce how to prepare the input for this function separately.

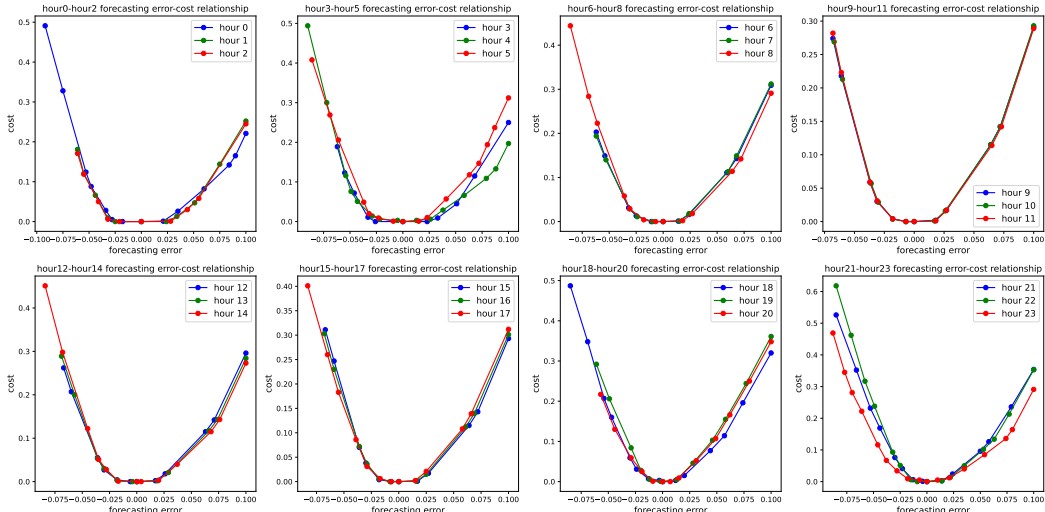

Figure 22: Loss function visualization.

```python
def calculate_scenario(data,
                       target,
                       methods_to_train,
                       horizon,
                       train_ratio,
                       feature_transformation,
                       time_stationarization,
                       datetime_features,
                       target_lag_selection,
                       external_feature_selection,
                       post_processing_quantile,
                       post_processing_value,
                       evaluation_metrics,
                       ):
                       '''
                       Code
                       '''
```

**data**  The load dataset, we need to input the data in Pandas format.

**target**  The column name of the variable we need to predict in the input Pandas, and other columns will be treated as external variables.

**methods_to_train**  A list containing the forecasting methods we need to include.

**horizon**  The default is to make predictions 24 hours in advance and multiple time scale forecasting can be made by adjusting this.

**train_ratio**  The division of the training and test sets.

**feature_transformation**  The strategies that used to stabilize the time series, including logarithmic transformation, data differentiation, and so on.

**time_stationarization**  The division of the training and test sets.

**datetime_features** Define what calendar variables to consider, such as day, month, year, whether it is holiday, etc.

**target_lag_selection** Define how to select historical data for forecasting. In our default settings and our benchmark, we will select values from the same time point in the past seven days to forecast the corresponding values for the eighth day. In addition, we also provide a strategy for selecting highly correlated historical data based on the autocorrelation of the data.

**external_feature_selection** We provide two strategies for selecting external variables: direct input and based on temperature calendar variable relationships.

**post_processing_quantile** Quantile-based forecasting may sometimes result in lower quantiles being greater than higher quantiles, and the main focus here is to rearrange them.

**post_processing_value** To limit the final output result, such as forcing the forecasting result to not exceed a certain value.

**evaluation_metrics** We include various evaluation metrics for users to choose from, which can be referenced specifically from D.2.

### C.2.2 HOW TO ADD NEW MODELS

Our framework mainly focuses on quantile-based probabilistic forecasting, and to add new models, we need to make definitions for the new models.

```python
class MYQuantile_Regressor(MultiQuantileRegressor):
    def __init__(self, quantiles: List[float] =
    ↪ [0.1,0.2,0.3,0.4,0.5,0.6,0.7,0.8,0.9]):
        super().__init__(
            X_scaler=StandardScaler(),
            y_scaler=StandardScaler(),
            quantiles=quantiles)

    def set_params(self, input_dim:
    ↪ int,external_features_diminsion: int):
        self.model = models.pytorch.PytorchRegressor(
            model=models.pytorch.MYQuantile_Model(input_dim,
            external_features_diminsion,
            n_output=len(self.quantiles)),
            loss_function=
            pytorchtools.PinballLoss(self.quantiles))
        return self
```

In addition, users also need to provide a specific details of the forecastng model, that is, how to handle the output of the model and ultimately convert it into output. Note that here we need to provide information on how the model handles external variables. Generally speaking, the specific form of external variables is related to the external variable processing strategy we define. Common external variables include meteorological factors such as temperature. If there are no external variables, we will use the corresponding calendar variables as external variables.

```python
class MYQuantile_Model(nn.Module):
    def __init__(self,input_parameters):
        super(MYQuantile_Model, self).__init__()
        '''
        build your model here
        '''
    def forward(self, X_batch,X_batch_ex):
        '''
        input the data into the model,
        here X_batch is the sequence data while
        X_batch_ex is the external variable.
        '''
        return output
```

## D  BENCHMARK EVALUATION

### D.1  HYPER PARAMETERS

In this section, we will introduce the hyperparameter settings in our load forecasting archive. Table 4 shows the parameter settings for non-deep learning methods. Here, BMQ represents the moving quantity method based on a fixed number of past time points, while BEQ is based on all historical data. BECP represents that the forecasting error obtained by the persistence method on the training set is directly added to the forecasting results obtained by the persistence method as quantile forecasting. QCE is similar but replaces the persistence method with linear regression. In addition, there are quantile regression methods based on the K-nearest neighbor algorithm (Hastie et al., 2009), quantile regression methods based on random forest and sample random forest (Meinshausen & Ridgeway, 2006), and quantile regression methods based on extreme random tree and sample extreme random tree (Geurts et al., 2006).

Table 4: Parameter settings for non-deep learning methods.

| Method | Parameters | | | |
|--------|-------------|-------------|-------------|-----------|
|        | Window size | N_neighbors | N_estimators | Quantiles |
| BMQ  | 7   | -  | -   | 0.01~0.99 |
| BEQ  | all | -  | -   | 0.01~0.99 |
| BCEP | -   | -  | -   | 0.01~0.99 |
| CE   | -   | -  | -   | 0.01~0.99 |
| KNNR | -   | 20 | -   | 0.01~0.99 |
| RFR  | -   | -  | 100 | 0.01~0.99 |
| SRFR | -   | -  | 100 | 0.01~0.99 |
| ERT  | -   | -  | 100 | 0.01~0.99 |
| SERT | -   | -  | 100 | 0.01~0.99 |

Apart from those, we introduce several deep learning methods and they are feed-forward neural networks(FFNN) (Jain et al., 1996), LSTM networks (Hochreiter & Schmidhuber, 1997) for sequence modeling, convolutional neural networks (Li et al., 2021), and Transformer (Vaswani et al., 2017) networks applying attention mechanisms. Additionally, we also have methods that modify the above neural network structures to make them more suitable for time series forecasting, such as LSTNet (Lai et al., 2018), which is designed to simultaneously capture both long-term and short-term patterns of time series, Wavenet based on causal convolution (Oord et al., 2016), and N-BEATS stacked into blocks using multiple linear layers (Oreshkin et al., 2020). Tables 5 and 6 respectively show

the hyperparameter settings of the training process and the network structure and parameters of the relevant deep learning methods. We divide the entire dataset into training and test sets at a ratio of 0.2, and then divide the training set into the final training and validation sets at a ratio of 0.2. To reduce the impact of neural network overfitting, we enable the early stop mechanism. Specifically, when the loss on the validation does not decrease for 15 epochs, we will stop training.

Table 5: Training process parameters.

| Parameters | | | | | |
|---|---|---|---|---|---|
| **Loss function** | **Validation ratio** | **Epochs** | **Patience** | **Optimizer** | **Learning rate** |
| MSE custom loss function PinballLoss(0.01∼0.99) | 0.2 | 1000 | 15 | torch.Adam | 0.0005 |

Table 6: Parameter settings for deep learning methods, here $e$ represents the dimension of the external variables.

| Method | Parameters | | |
|---|---|---|---|
| | **Network structure** | **Network parameters** | **Quantiles** |
| FFNN | 2 Dense layers | Dense_1: (1+$e$,50) Dense_2: (50,1) | 0.01∼0.99 |
| LSTM | 1 LSTM layer 1 Dense layer | LSTM: (1,64,2) Dense: (1+$e$,1) | 0.01∼0.99 |
| CNN | 3 Conv1d layers 2 Maxpool1d layers 2 Dense layers | Conv1d_1: (1+$e$, batch size, 1, 1) Conv1d_2: (batch size, 128, 1, 3) Conv1d_3: (128, 256, 1, 3) Dense_1: (256, 128) Dense_2: (128, 1) | 0.01∼0.99 |
| Transformer | 1 Positional Encoding layer 1 Encoding layer 1 Decoding layer 1 Transformer layer 1 Dense layer | Encoding: (1, 256) Decoding: (1, 256) Transformer: d_model=256, n_head=4, dim_forward=512 Dense: (7×256+$e$, 1) | 0.01∼0.99 |
| LSTNet | 1 Conv2d layer 2 GRU layers 2 hidden(Dense) layers 1 Dense layer | Conv2d: (1, 16, 2, 1) GRU_1: (16, 16) GRU_2: (16, 32) hidden_1: (16+1×32, 1) hidden_2: (16, 1) Dense: (1+$e$, 1) skip=1 highway=7 | 0.01∼0.99 |
| WaveNet | 1 CausalConv1d layer 1 DilatedStack 2 Conv1d layers 1 Dense layer | CausalConv1d: (1, 16, 2, 1) DilatedStack: residual size=16 skip size=4 dilation depth=2 Conv1d_1: (4, 1, 1, 0) Conv1d_2: (1, 1, 1, 0) Dense: (1+$e$, 1) | 0.01∼0.99 |
| N-BEATS | 1 Trend Stack 1 Seasonal Stack 1 Dense layer | Trend: hidden=64, theta dim=(4,8) Seasonal: hidden=64, theta dim=(4,8) Dense: (1+$e$,1) | 0.01∼0.99 |

### D.2 EVALUATION METRICS

To evaluate the forecasting performance of different methods in our set day-ahead forecasting, we will introduce many evaluation metrics, which are divided into metrics for point forecasting and metrics for probabilistic forecasting. It is worth noting that not all metrics are used to directly distinguish forecasting performance, and some of them may be used to describe the shape of probabilistic forecasting, thereby more comprehensively presenting the forecasting characteristics of different models. We will provide a detailed introduction below.

#### D.2.1 POINT FORECASTING EVALUATION

Similar to (Godahewa et al., 2021), we adopt **4 metrics** that are widely used to evaluate the results of deterministic forecasting, and they are MAPE (Mean Absolute Percentage Error), MASE (Mean Absolute Scaled Error), RMSE (Root Mean Squared Error), and MAE (Root Mean Squared Error) respectively. Their mathematical definitions are listed below, note that $\{y_t\}_{t=1}^n$ represents the actual value and $\{F_t\}_{t=1}^n$ represents the predicted one.

- **MAPE**. MAPE is a metric of forecasting accuracy that calculates the average percentage of forecasting error for all data points. The smaller the value of MAPE, the higher the forecasting accuracy. Due to its percentage error, it can be used to compare forecasting performance at different scales. However, MAPE may result in an infinite or very large error percentage for zero or near zero actual values. The formal definition of MAPE is given below

$$\text{MAPE} = \frac{1}{n} \sum_{t=1}^n \left| \frac{y_t - F_t}{y_t} \right| \times 100\%.$$

- **MASE**. MASE is a scale-independent error measure that calculates errors by comparing the forecasting error with the average absolute first-order difference of the actual value sequence. The advantage of MASE is that it is not affected by the size of actual values, so it is more robust for forecasting problems of different sizes. The formal definition of MASE is given below

$$\text{MASE} = \frac{1}{n} \sum_{t=1}^n \frac{|y_t - F_t|}{\frac{1}{n-1} \sum_{t=1}^n |y_t - y_{t-1}|}.$$

- **RMSE**. RMSE is a commonly used measure of forecasting error that calculates the square root of the average of the sum of squares of forecasting errors for all data points. The smaller the value of RMSE, the higher the forecasting accuracy. It is sensitive to outliers, which may lead to large forecasting errors. However, RMSE has good interpretability because its units are the same as the actual and predicted values. The formal definition of RMSE is given below

$$\text{RMSE} = \sqrt{\frac{1}{n} \sum_{t=1}^n (y_t - F_t)^2}.$$

- **MAE**. MAE is also a commonly used measure of forecasting error, which calculates the average of the absolute value of forecasting errors for all data points. The smaller the value of MAE, the higher the forecasting accuracy. Compared with RMSE, MAE is less sensitive to outliers, so it may be more robust in the case of outliers. The formal definition of MAE is given below

$$\text{MAE} = \frac{1}{n} \sum_{t=1}^n |y_t - F_t|.$$

#### D.2.2 PROBABILISTIC FORECASTING EVALUATION

Compared to point forecasting, probabilistic forecasting can provide more information. Therefore, we can evaluate the results of probabilistic forecasting from more aspects. We have summarized a total of **11 metrics** to comprehensively evaluate the load probabilistic forecasting results and list them below. Note that we will also perform matrix visualization based on some metrics to help users better evaluate different prediction models.

- **CoverageError (CE)**. CoverageError is a method of measuring the quality of forecasting intervals, which measures the difference between the proportion of actual observations falling within the forecasting interval and the expected coverage rate. A smaller CoverageError indicates that the forecasting interval captures actual observations more accurately. Here, $L_t$ and $U_t$ represent the lower and upper bound of the forecasting interval while $UB$ and $LB$ respectively represent the upper and lower bounds of the interval we want. It is worth noting that when we visualize it, we call it ReliabilityMatrix. Specifically, we first divide the quantiles into the upper half and the lower half with 0.5 as the boundary. And perform pairwise combinations to obtain different nominal coverage rates as the horizontal axis, while the vertical axis represents the actual coverage rate.

$$CE = \frac{1}{n} \sum_{t=1}^{n} (I(L_t \leq y_t \leq U_t) - (UB - LB)).$$

- **Winkler Score (WS)**. Winkler Score (WS) is a metric that measures the quality of forecasting intervals. The forecasting interval is the forecasting range for future observations, usually represented by a lower bound and an upper bound. Winkler Score is used to evaluate whether the forecasting interval accurately captures actual observations, taking into account the width of the interval. A lower Winkler Score indicates better forecasting interval quality. Here, the symbols used are the same as CE while $\delta = U_t - L_t$. Similar to CE, in the corresponding visualization matrix, the abscissa should be different nominal coverage rates, and for a central (1-$\alpha$ )% forecasting interval, it is defined as follows:

$$WS_{a,t} = \begin{cases} \delta, & L_t \leq y_t \leq U_t. \\ \delta + \frac{2(y_t - U_t)}{\alpha}, & y_t > U_t. \\ \delta + \frac{2(L_t - y_t)}{\alpha}, & y_t < L_t. \end{cases}$$

- **Pinball Loss (PL)**. Pinball Loss considers the difference between the forecasting value and the actual observation value, and weights the error based on whether the forecasting value falls on the side of the actual observation value (above or below). This enables Pinball Loss to capture the uncertainty in probabilistic forecasting and assign different weights to symmetric errors in loss calculations. A lower Pinball Loss indicates a smaller error between probabilistic forecasting and actual observations. Here, $L_\tau$ represents the Pinball Loss at the quantile $\tau$ and $\hat{y}_{\tau,t}$ is the forecasting value of corresponding time and quantile. In our setting, we consider the sum of 99 quantiles from 0.01 to 0.99, and it is defined as follows:

$$PL = \frac{1}{n_\tau \cdot n} \sum_{t=1}^{n} \sum_{i=1}^{n_\tau} L_\tau \left( \hat{y}_{\tau,t}, y_t \right).$$

- **RampScore (RS)**. RampScore measures the consistency of the slope (i.e. increasing or decreasing trend) between the forecasting sequence and the actual observation sequence. Firstly, we use the Swing Door compression algorithm (Khan et al., 2020) to compress the forecasting sequence and the observed sequence, and then calculate the first-order difference values of these two sequences separately. Finally, we calculate the absolute difference between the first-order difference values of the two sequences and take the average to obtain the RampScore. A lower RampScore indicates that the model is more capable of capturing trends in sequence changes. Here, we calculate RampScore for 9 quantiles from 0.1 to 0.9.
- **CalibrationError**. CalibrationError (Chung et al., 2021) mainly evaluates the accuracy of forecasting models in representing uncertainty. The CalibrationError represents the difference between the forecasting quantile and the actual quantile. A smaller CalibrationError means that the forecasting model has higher accuracy in representing uncertainty, while a larger calibration error means that the forecasting model has lower accuracy in representing uncertainty. In the visualization matrix, we show the proportion of the predicted value greater than the true value under different quantiles. The closer the forecasting method is to the line y=x, the better the performance will be.

In addition to the metrics mentioned above, we also provide many other metrics. Although we will not present each of them in detail here, interested users can easily visualize them with the open-source code we provide. These evaluation metrics include IntervalWidth, QuantileCrossing, BoundaryCrossing, Skewness, Kurtosis, and QuartileDispersion. Among them, IntervalWidth calculates the width

of probabilistic forecasting intervals given by different methods while QuantileCrossing gives the ratio of any two quantiles in which the predicted value of the lower quantile is greater than the predicted value of the higher quantile. BoundaryCrossing calculates the probability that the true value falls outside the forecasting range. Skewness and Kurtosis are metrics that describe the shape of a probability distribution. As for QuartileDispersion, its detailed description can be found in (Bonett, 2006).

## D.3 Evaluation results

In this section, we will mainly demonstrate the forecasting performance of 14 out of the 16 probabilistic forecasting methods we mentioned earlier, as well as 7 point forecasting methods. The relevant results of the two methods based on moving average can be obtained in the repository we provide.

### D.3.1 Running time

In our archive, all experiments were conducted on Intel (R) Xeon (R) W-3335 CPU @ 3.40GHz and NVIDIA GeForce RTX3080Ti. Table 7, Table 8, Table 9, and Table 10 report the training and inference time of various methods separately (note that the time for calculating metrics is not included). From the perspective of deep learning probabilistic forecasting models, incorporating temperature feature engineering not only improves the forecasting performance (see section 5.1 and Appendix D.5) but also reduces the time spent. For non-deep learning models, incorporating temperature feature engineering greatly increases the required time. This may be because our feature engineering incorporates a large amount of sparse data, which is difficult for nondeep learning models to handle. From the perspective of point forecasting, the traditional MSE loss and asymmetric loss functions take approximately the same amount of time.

Table 7: Comparison of running time for probabilistic forecasting (except for ELF and UCI).

(a) Deep learning methods

| Time(s) | FFNN | CNN | LSTM | LSTN | WaveNet | NBEATS | Transformer |
|---|---|---|---|---|---|---|---|
| With feature engineering | 671.898 | 668.576 | 1565.256 | 2396.536 | 3665.051 | 3915.706 | 6484.478 |
| Without feature engineering | 1107.768 | 608.697 | 1738.488 | 4651.474 | 4717.738 | 5292.745 | 5097.802 |

(b) Non-deep learning methods

| Time(s) | BMQ | BEQ | BCEP | CE | KNNR | RFR | SRFR | ERT | SERT |
|---|---|---|---|---|---|---|---|---|---|
| With feature engineering | 46.436 | 109.675 | 108.340 | 60.916 | 70.052 | 4548.035 | 3967.662 | 4542.447 | 4538.864 |
| Without feature engineering | - | - | 108.395 | 0.934 | 60.196 | 1861.025 | 1318.141 | 1276.955 | 1315.708 |

Table 8: Comparison of running time for probabilistic forecasting (for ELF and UCI).

(a) Deep learning methods

| Time(s) | FFNN | CNN | LSTM | LSTN | WaveNet | NBEATS | Transformer |
|---|---|---|---|---|---|---|---|
| Without feature engineering | 6465.294 | 3187.983 | 8666.618 | 21245.597 | 23781.845 | 31907.537 | 26157.561 |

(b) Non-deep learning methods

| Time(s) | BMQ | BEQ | BCEP | CE | KNNR | RFR | SRFR | ERT | SERT |
|---|---|---|---|---|---|---|---|---|---|
| Without feature engineering | 180.455 | 444.142 | 425.788 | 3.386 | 145.617 | 6151.749 | 4809.492 | 4256.425 | 5129.695 |

## D.4 Point forecasting results

Tables 11, 12, and 13 respectively report the performance comparison of forecasting models based on the MSE loss function in several different datasets. Among them, only a portion of the UCI dataset is

Table 9: Comparison of running time for different loss functions (except for ELF and UCI).

| Time(s) | FFNN | CNN | LSTM | LSTN | WaveNet | NBEATS | Transformer |
|---|---|---|---|---|---|---|---|
| MSE without feature engineering | 1116.204 | 545.852 | 1145.189 | 2918.964 | 2869.714 | 4043.892 | 4475.159 |
| MSE with feature engineering | 585.367 | 716.680 | 1388.907 | 2369.784 | 3680.179 | 3606.027 | 6356.714 |
| custom loss function with feature engineering | 1275.936 | 1146.166 | 2106.425 | 3439.587 | 4615.414 | 4400.446 | 7074.623 |

Table 10: Comparison of running time for different loss functions (for ELF and UCI).

| Time(s) | FFNN | CNN | LSTM | LSTN | WaveNet | NBEATS | Transformer |
|---|---|---|---|---|---|---|---|
| MSE | 5216.818 | 2386.388 | 7562.510 | 15905.827 | 15621.957 | 23684.826 | 20519.646 |
| custom loss function | 12066.808 | 4485.706 | 14169.861 | 23616.685 | 21856.217 | 29754.229 | 23146.784 |

reported. The remaining results and other evaluation metrics can be found in the Github[2]. From the perspective of MAPE metrics, in some datasets, existing forecasting models cannot provide reasonable forecasting results. This situation is particularly severe in building-level datasets. Fortunately, in the aggregated level data, the vast majority of prediction models can provide reasonable prediction results. This is due to the stronger stationarity of aggregated level data compared to the building-level ones. From the perspective of predictive models, models based on simple structures perform better than relatively complex models such as LSTNet, WaveNet, and N-BEATS. Among them, WaveNet performs the worst in multiple datasets, indicating that it is not suitable for application in the scenario of day-ahead forecasting of the power grid.

### D.5 PROBABILISTIC FORECASTING RESULTS

Similar to point forecasting, we present the PinballLoss results of the forecasting model on some datasets in Tables 14, 15, 16, and 17, while placing other results in repository for users to read. These tables present the forecasting results for all datasets with temperature variables.

- From the perspective of forecasting models, the performance of non-deep learning forecasting models is generally better than that of deep learning models. Among them, ERT and SERT perform relatively well. In deep learning methods, LSTM networks perform well while Transformer, FFNN, and CNN can also achieve the best results on certain datasets. However, the relatively complex neural network methods are lagging on the vast majority of datasets, which is relatively consistent with the results of point forecasting methods.
- From the perspective of feature engineering, when there is no one-to-one correspondence between temperature data and load data in some data sets, such as Spain and GEF12, feature engineering on temperature may reduce forecasting accuracy. When there is a clear one-to-one correspondence, such as GEF17, feature engineering of temperature-calendar variables will greatly improve the model's forecasting performance.

Figure 23, 24 report the evaluation result of two data in the GEF17 dataset respectively as an illustration. These three figures show the results after adding the temperature-calendar variable feature engineering. In most of these figures, the abscissa represents different quantiles, and the ordinate is the corresponding evaluation metric. The best-performing methods on these three datasets by the PinballLoss metric are FFNN, CNN, and QCE. Take CT as an example. Although FFNN performs best in PinballLoss, it cannot maintain an advantage in other metrics. For WinklerScoreMatrix, the LSTM model performs better than the FFNN method under high nominal coverage. This indicates that the LSTM model is superior to FFNN in considering extreme scenarios. For RampScoreMatrix, although FFNN has achieved good results in low quantiles, it is not better than the simple QCE method in high quantiles. Similar results have also been observed in other datasets and it is rare for a forecasting method to overwhelm all other forecasting methods in all aspects. These examples show that different quantiles can be considered separately as well as different metrics are needed to focus

---

[2]https://anonymous.4open.science/r/ProEnFo-17CC

on different aspects of the forecasting model so that different forecasting models can be distinguished. And this is also a major contribution of our load forecasting archive.

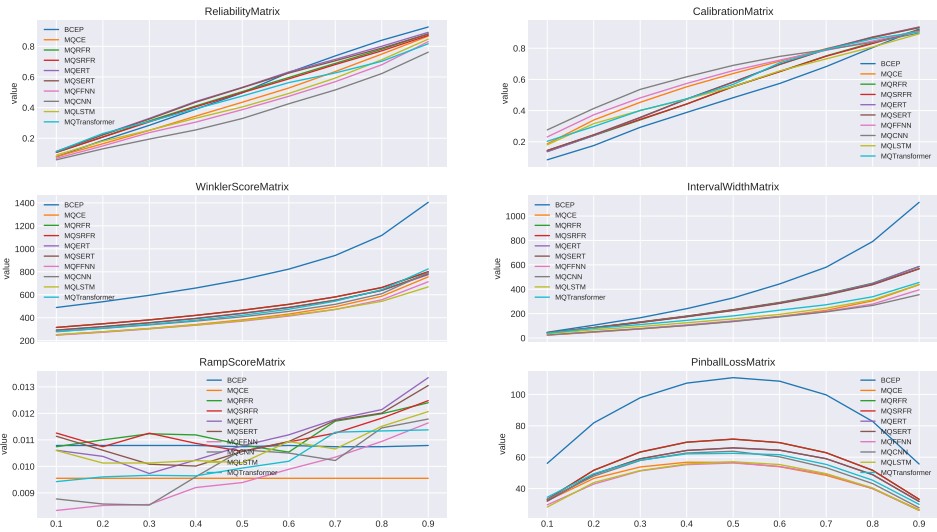

Figure 23: Visualization evaluation metrics in the GEF17 CT dataset.

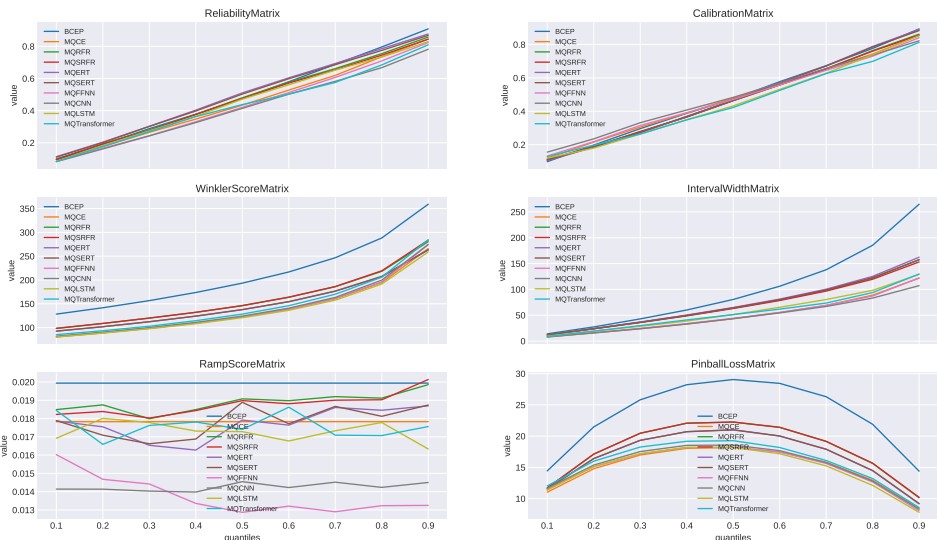

Figure 24: Visualization evaluation metrics in the GEF17 ME dataset.

# E    DOCUMENTATION

**Long-term preserve plan**: Currently, our relevant datasets and prediction results are saved in the folder in a cloud service. This is mainly because we are still updating it, and the main direction is to add more fine-grained data related to smart meters. After our dataset is fully developed, we will apply for the relevant DOI for it.

**Author statement**: We confirm that the relevant dataset sources comply with relevant regulations and we bear all responsibility in case of violation of rights, etc., and confirmation of the data license.

Table 11: MAPE comparison results based on MSE Loss function in datasets with temperature variables (I), the coloring indicates that the current forecasting model cannot obtain reasonable results on this dataset.

| MAPE(%) | Methods | | | | | | |
|---|---|---|---|---|---|---|---|
| | FFNN | LSTM | CNN | LSTNet | WaveNet | N-BEATS | Transformer |
| GEF12_1 | 9.21 | 9.53 | **9.16** | 9.32 | 9.77 | 11.72 | 9.73 |
| GEF12_2 | 5.58 | **5.41** | 5.53 | 5.57 | 6.99 | 7.03 | 5.49 |
| GEF12_3 | 5.65 | 5.48 | 5.59 | **5.46** | 7.02 | 6.97 | 5.51 |
| GEF12_4 | 27.36 | **26.28** | 26.67 | 26.8 | 28.4 | 27.04 | 26.96 |
| GEF12_5 | 9.3 | 9.19 | **9.17** | 9.47 | 11.05 | 9.52 | 9.62 |
| GEF12_6 | 5.53 | **5.48** | 5.65 | 5.53 | 6.89 | 5.57 | 5.57 |
| GEF12_7 | 5.65 | 5.48 | 5.59 | **5.46** | 7.02 | 6.97 | 5.51 |
| GEF12_8 | 8.48 | 8.08 | 8.12 | 8.45 | 11.16 | 8.16 | **8.02** |
| GEF12_9 | 118.52 | 107.78 | 124.49 | 122.08 | 141.54 | 140.2 | 99.19 |
| GEF12_10 | **26.21** | 31 | 27.29 | 26.81 | 39.4 | 33.38 | 31.74 |
| GEF12_11 | 6.87 | 6.97 | 6.8 | 7.01 | 10.08 | 6.6 | **6.38** |
| GEF12_12 | 7 | 7.17 | **6.46** | 7.36 | 10.03 | 7.16 | 7.1 |
| GEF12_13 | **7.67** | 7.67 | 7.96 | 7.76 | 9.67 | 7.74 | 7.88 |
| GEF12_14 | **10.57** | 10.97 | 10.58 | 10.86 | 13.29 | 11.77 | 11.08 |
| GEF12_15 | 8.32 | **8.21** | 8.29 | 8.36 | 10.61 | 8.88 | 8.67 |
| GEF12_16 | **8.46** | 8.88 | 8.54 | 8.84 | 10.26 | 9.82 | 9.21 |
| GEF12_17 | 7.26 | 7.01 | **6.79** | 7.08 | 9.76 | 7.41 | 7.01 |
| GEF12_18 | 8.21 | 8.27 | **8.14** | 8.28 | 10.35 | 8.67 | 8.32 |
| GEF12_19 | 9.85 | 9.83 | **9.56** | 9.78 | 12.66 | 10.16 | 9.98 |
| GEF12_20 | 6.8 | 6.44 | 6.64 | **6.39** | 9.36 | 9.4 | 6.5 |
| GEF14 | **2.26** | 2.48 | 2.52 | 2.36 | 3.02 | 3.01 | 2.44 |
| GEF17_CT | 3.76 | 3.75 | **3.63** | 3.78 | 5.02 | 4 | 4.29 |
| GEF17_ME | 3.08 | 3.08 | 3.21 | 3.07 | 3.2 | 3.22 | **3.06** |
| GEF17_NEMASSBOST | 3.85 | **3.51** | 3.84 | 3.69 | 4.67 | 3.98 | 3.88 |
| GEF17_NH | 3.67 | 3.7 | **3.66** | 4.76 | 4.61 | 3.85 | 3.93 |
| GEF17_RI | **3.3** | 3.38 | 3.34 | 3.36 | 4.4 | 4.48 | 3.55 |
| GEF17_SEMASS | **4.76** | 4.84 | 4.8 | 4.85 | 5.58 | 5.17 | 5.07 |
| GEF17_VT | 5.01 | **4.96** | 5.03 | 5 | 6.2 | 6.11 | 5.55 |
| GEF17_WCMASS | 3.88 | 3.8 | 3.8 | **3.7** | 4.86 | 4.11 | 3.92 |
| Covid19 | 5.19 | 5.16 | 5.55 | 6.54 | 11.71 | 11.71 | **4.9** |
| Bull_assembly_Amalia | 21.92 | **20.96** | 21.02 | 21.53 | 24.07 | 23.86 | 21.19 |
| Bull_assembly_Goldie | 20.34 | 18.49 | **18.27** | 23.78 | 24.79 | 19.79 | 20.12 |
| Bull_assembly_Lance | **26.42** | 26.6 | 26.43 | 27.26 | 27.7 | 27.38 | 27.27 |
| Bull_assembly_Maren | 86.83 | 60.86 | 89.6 | 111.2 | 124.41 | 77.62 | 53.34 |
| Bull_assembly_Nathanial | 12.12 | 11.67 | **10.88** | 11.19 | 12.68 | 13.67 | 11.42 |
| Bull_assembly_Vanessa | 21.52 | 20.12 | 21.34 | **19.4** | 23.4 | 19.88 | 20.22 |
| Bull_education_Arthur | 34.48 | 31.22 | 33.86 | 32.37 | 43.8 | 31.51 | **30.82** |
| Bull_education_Bernice | 21.11 | **20.13** | 20.99 | 20.24 | 26.06 | 22.33 | 20.74 |
| Bull_education_Brain | 29.12 | 27.88 | **27.5** | 29.52 | 31.81 | 30.89 | 27.53 |
| Bull_education_Brenda | 39.39 | 38.99 | **35.22** | 36.29 | 43.16 | 36.07 | 39.96 |
| Bull_education_Dan | 22.79 | 21.89 | **21.46** | 23.88 | 29.66 | 22.06 | 21.82 |
| Bull_education_Dora | 31.19 | 29.44 | **27.67** | 29.52 | 29.8 | 30.77 | 29.65 |
| Bull_education_Dottie | 33.92 | 21.67 | 30.64 | 20.37 | 45.3 | **19.78** | 19.93 |
| Bull_education_Elva | 37.03 | 39.39 | **34.51** | 38.09 | 43.1 | 40.74 | 36.9 |
| Bull_education_Gregory | 127.84 | 52.35 | 148.31 | 85.61 | 84.84 | 429.83 | 36.06 |
| Bull_education_Jae | 14.36 | 14.55 | 14.92 | **14.13** | 20.3 | 18.81 | 15.57 |
| Bull_education_Joseph | **22.98** | 24.14 | 23.07 | 23.27 | 31.05 | 24.45 | 26.48 |
| Bull_education_Kendra | 20.69 | 21.31 | 20.71 | 21.71 | 35.31 | 21.01 | **19.91** |
| Bull_education_Krista | 65.97 | 62.57 | **60.66** | 64.12 | 85.78 | 86.32 | 63.59 |
| Bull_education_Lenny | 34.1 | **29.99** | 31.52 | 31.53 | 47.07 | 30.59 | 30.66 |
| Bull_education_Linnie | 31.24 | **29.19** | 31.6 | 35.97 | 36.4 | 29.77 | 29.62 |

Table 12: MAPE comparison results based on MSE Loss function in datasets with temperature variables (II), the coloring indicates that the current forecasting model cannot obtain reasonable results on this dataset.

| MAPE(%) | Methods | | | | | | |
|---|---|---|---|---|---|---|---|
| | FFNN | LSTM | CNN | LSTNet | WaveNet | N-BEATS | Transformer |
| Bull_education_Luke | 22.73 | **22.23** | 22.68 | 22.9 | 23.75 | 22.48 | 22.28 |
| Bull_education_Magaret | 137.24 | 108.8 | 138.13 | 109.81 | 190.14 | 109.75 | 109.24 |
| Bull_education_Mervin | 37.97 | 33.91 | **33.68** | 34.04 | 34.33 | 47.01 | 39.5 |
| Bull_education_Miranda | 21.35 | 20.8 | 21.38 | 21.35 | 25.53 | **20.36** | 20.38 |
| Bull_education_Patrina | 30.02 | 29.06 | 30 | **28.97** | 34.97 | 29.27 | 29.82 |
| Bull_education_Racheal | 56.78 | 65.22 | 58.9 | 64.3 | **55.41** | 67.88 | 68.37 |
| Bull_education_Reina | 30.04 | 28.79 | **25.6** | 26.84 | 32.55 | 28.89 | 26.5 |
| Bull_education_Roland | 378.96 | 418.75 | 356.88 | 375.33 | 381.9 | 478.95 | 467.88 |
| Bull_education_Roseann | 45.68 | 43.38 | 44.55 | 43.97 | 46.57 | 45.81 | **42.81** |
| Bull_lodging_Carie | 107.47 | 98.5 | 112.22 | 103.09 | 146.3 | 84.83 | 101.97 |
| Bull_lodging_Hugo | 55.93 | **53.55** | 70.42 | 59.78 | 65.33 | 55.56 | 61.16 |
| Bull_lodging_Jeremiah | 13.18 | **12.22** | 13.9 | 14.38 | 21.95 | 14.73 | 13.43 |
| Bull_lodging_Lettie | 25.19 | 25.65 | 24.71 | **24.7** | 29.7 | 25.14 | 27.04 |
| Bull_lodging_Melissa | 20.8 | 23.35 | **20.77** | 22.64 | 24.37 | 22.49 | 25.61 |
| Bull_lodging_Perry | 25.13 | 24.1 | **23.45** | 24.54 | 24.15 | 26.96 | 28.28 |
| Bull_lodging_Terence | 24.65 | 22.94 | 27.15 | **22.26** | 32.09 | 22.7 | 22.67 |
| Bull_lodging_Travis | 15.37 | 15.3 | 17.97 | 22.81 | 31.7 | 14.67 | **14.25** |
| Bull_office_Chantel | 78.93 | 68.54 | 79.19 | 73.42 | 96.3 | **68.03** | 68.57 |
| Bull_office_Yvonne | 29.72 | 23.36 | 34.93 | 25.04 | 48.33 | **22.8** | 23.29 |
| Bull_public_Hyun | 21.22 | 20.7 | **19.61** | 21.09 | 22.19 | 21.48 | 23.34 |
| Hog_education_Haywood | 16.44 | 16.58 | **12.97** | 20.98 | 19.96 | 16.01 | 17.47 |
| Hog_education_Jewel | 22.53 | 22.04 | 23.67 | 31.47 | 30.49 | 21.26 | **20.21** |
| Hog_education_Leandro | 120.88 | 234.08 | 211.67 | 182.21 | 241.2 | 192.01 | 208.26 |
| Hog_education_Luvenia | 21.09 | 18.64 | **16.75** | 18.26 | 20.38 | 18.4 | 19.44 |
| Hog_education_Sonia | 186.16 | 224.47 | 214.12 | 223.01 | 319.63 | 205.18 | 279.88 |
| Hog_lodging_Shanti | 106.31 | 81.9 | 101.81 | 88.25 | 161.28 | 158.54 | 75.17 |
| Hog_office_Betsy | **16.82** | 23.57 | 18 | 21.91 | 21.6 | 17.92 | 18.74 |
| Hog_office_Byron | 30.03 | 27.33 | **25.12** | 27.51 | 49.41 | 49.59 | 27.7 |
| Hog_office_Candi | 169.49 | 167.73 | 179.04 | 173.45 | 322.46 | 175.25 | 164.42 |
| Hog_office_Charla | 11.72 | 12.44 | **11.61** | 15.07 | 14.74 | 14.91 | 12.48 |
| Hog_office_Corey | 58.37 | 72.63 | **56.7** | 70.6 | 76.69 | 78.21 | 76.82 |
| Hog_office_Cornell | 22.39 | 19.87 | 25.47 | 20.82 | 20.82 | 30 | **19.27** |
| Hog_office_Elizbeth | **14.07** | 16.62 | 17.09 | 15.37 | 21.97 | 15.78 | 16.19 |
| Hog_office_Elnora | 16.56 | 16 | **14.87** | 19.62 | 19.73 | 20.04 | 16.43 |
| Hog_office_Leanne | 78.77 | **47.73** | 78.73 | 72.48 | 90.79 | 92.55 | 71.63 |
| Hog_office_Nia | 25.21 | 23.6 | 23.77 | **23.44** | 29.73 | 32.57 | 25.06 |
| Hog_office_Richelle | 100.88 | 73.73 | 96.45 | 74.58 | 74.87 | 63.8 | 74.83 |
| Hog_office_Roger | 8.05 | 7.67 | 8.47 | **6.76** | 13.67 | 9.05 | 7.54 |
| Hog_office_Rolando | 15.16 | 18.08 | **14.55** | 18.63 | 21.87 | 16.8 | 17.2 |
| Hog_office_Terry | 12.29 | 12.34 | 12.81 | 12.06 | 25.49 | 12.24 | **11.19** |
| Hog_public_Brad | **11.16** | 11.6 | 12.25 | 12.89 | 17.74 | 20.97 | 11.67 |
| Hog_public_Crystal | 450.2 | 280.87 | 426.0 | 274.48 | 291.52 | 592.19 | 281.31 |
| Hog_public_Kevin | 227.29 | 258.55 | 205.92 | 223.34 | 250.82 | 227.99 | 238.8 |
| Hog_public_Octavia | 19.01 | **17.12** | 19.94 | 17.96 | 21.71 | 21.15 | 17.67 |
| Cockatoo | 26.39 | 25.64 | **24.64** | 26.93 | 26.12 | 27.69 | 26.07 |
| PDB | 6.35 | 5.24 | 5.34 | 5.35 | 6.04 | **4.75** | 5.02 |
| Spain | 6.05 | 5.83 | 6 | 5.58 | 6.94 | **5.55** | 5.63 |

Table 13: MAPE comparison results based on MSE Loss function in datasets without temperature variables (Partial), the coloring indicates that the current forecasting model cannot obtain reasonable results on this dataset.

| MAPE(%) | Methods | | | | | | |
|---|---|---|---|---|---|---|---|
| | FFNN | LSTM | CNN | LSTNet | WaveNet | N-BEATS | Transformer |
| ELF_load | **4.07** | 4.26 | 4.07 | 4.38 | 4.64 | 4.39 | 4.33 |
| UCI_0 | 85.93 | 83.47 | 68.91 | 96.69 | 90.91 | 253.25 | 89.81 |
| UCI_1 | 5.52 | 5.36 | 5.55 | 5.41 | 18.3 | 6.55 | **5.35** |
| UCI_2 | 27.81 | 15.52 | 19.39 | 186.33 | 176.38 | 17 | 18.9 |
| UCI_3 | **7.1** | 7.47 | 7.25 | 7.26 | 24.75 | 8.07 | 7.45 |
| UCI_4 | **11.5** | 11.89 | 12.08 | 38.47 | 26.91 | 12.76 | 11.74 |
| UCI_5 | **6.92** | 7.32 | 6.95 | 7.23 | 24.88 | 8.24 | 7.66 |
| UCI_6 | 36.15 | 36.92 | 40.84 | **32.01** | 72.87 | 42.37 | 41.16 |
| UCI_7 | **5.93** | 6.18 | 6.17 | 6.17 | 14.55 | 6.64 | 6.24 |
| UCI_8 | **13.51** | 14.43 | 13.6 | 15.03 | 14.94 | 28.97 | 14.02 |
| UCI_9 | 23.01 | **22.4** | 22.88 | 23.15 | 43.88 | 23.65 | 24.65 |
| UCI_10 | **8.27** | 8.48 | 8.47 | 8.51 | 21.28 | 21.32 | 8.37 |
| UCI_11 | **13.31** | 13.48 | 13.48 | 14.54 | 13.9 | 13.65 | 13.52 |
| UCI_12 | **7.64** | 7.8 | 8.07 | 23.69 | 24.29 | 10.35 | 8.31 |
| UCI_13 | **8.77** | 8.97 | 9.09 | 8.94 | 25.66 | 11.36 | 10.21 |
| UCI_14 | 5.66 | 5.91 | **5.56** | 5.81 | 16.87 | 6.35 | 5.84 |
| UCI_15 | 11.35 | 11.44 | **11.29** | 11.61 | 23.52 | 23.63 | 11.61 |
| UCI_16 | 9.79 | **9.46** | 9.81 | 19.3 | 19.31 | 10.08 | 9.47 |
| UCI_17 | **12.35** | 12.47 | 12.42 | 12.5 | 12.5 | 18.71 | 12.51 |
| UCI_18 | **5.83** | 6.02 | 5.86 | 6.09 | 21.16 | 6.66 | 6.54 |
| UCI_19 | **12.78** | 13.11 | 12.79 | 13.94 | 37.44 | 18.66 | 13.5 |
| UCI_20 | 11.32 | 11.76 | **11.23** | 26.05 | 12.08 | 26.04 | 11.79 |
| UCI_21 | 9.97 | 9.97 | **9.79** | 10.07 | 31.48 | 31.58 | 10.14 |
| UCI_22 | 8.57 | 8.58 | **8.33** | 8.67 | 25.04 | 9.25 | 8.79 |
| UCI_23 | **9.02** | 9.11 | 9.14 | 33.44 | 27.28 | 23.81 | 9.22 |
| UCI_24 | **7.25** | 7.42 | 7.56 | 7.53 | 20.24 | 20.14 | 7.51 |
| UCI_25 | **14.35** | 14.78 | 14.42 | 15.63 | 51.78 | 14.38 | 16.06 |
| UCI_26 | 6.91 | 7.12 | **6.88** | 21.33 | 21.34 | 7.76 | 7.13 |
| UCI_27 | 8.79 | 8.88 | 8.88 | **8.71** | 22.93 | 22.93 | 8.92 |
| UCI_28 | 9 | **8.97** | 9.16 | 9.18 | 18.54 | 9.94 | 9.48 |
| UCI_29 | 38.46 | 40.07 | 37.95 | 37.85 | 101.18 | 49.83 | 39.39 |
| UCI_30 | 9.94 | **9.72** | 10.49 | 10.04 | 23.17 | 10.78 | 9.9 |
| UCI_31 | 5.82 | 5.91 | 5.81 | **5.79** | 18.68 | 6.17 | 5.93 |
| UCI_32 | 41.21 | 67.08 | 23.04 | 113.77 | 362.63 | 47.05 | 89.83 |
| UCI_33 | **7.98** | 8.08 | 8.16 | 15.86 | 8.15 | 8.31 | 8.32 |
| UCI_34 | **6.71** | 6.95 | 6.77 | 7.21 | 7.03 | 7.76 | 7 |
| UCI_35 | **7.1** | 7.32 | 7.16 | 19.73 | 19.75 | 19.72 | 7.42 |
| UCI_36 | **21.95** | 22.8 | 22.11 | 23.09 | 63.61 | 24.83 | 22.97 |
| UCI_37 | **8.72** | 8.79 | 8.77 | 9.01 | 22.42 | 9.61 | 8.83 |
| UCI_38 | **9.34** | 9.71 | 9.48 | 9.73 | 21.71 | 10.4 | 9.65 |
| UCI_39 | **11.2** | 11.22 | 11.56 | 11.54 | 21.86 | 11.76 | 11.33 |
| UCI_40 | 6.28 | 6.45 | **6.25** | 6.56 | 18.2 | 18.1 | 6.62 |
| UCI_41 | 7.75 | 7.86 | **7.48** | 8.28 | 8.44 | 25.07 | 8.1 |
| UCI_42 | **10.75** | 10.8 | 10.9 | 10.87 | 22.12 | 11.37 | 10.84 |
| UCI_43 | 8.16 | **8.07** | 8.13 | 8.12 | 23.95 | 24.04 | 8.17 |
| UCI_44 | 8.98 | 9.37 | **8.88** | 9.24 | 31.73 | 9.78 | 9.32 |
| UCI_45 | **5.4** | 5.54 | 5.56 | 5.4 | 14.03 | 5.68 | 5.53 |
| UCI_46 | 15.56 | 13.98 | 15.85 | 36.2 | 33.63 | 43.56 | **13.57** |
| UCI_47 | **5.87** | 6.03 | 6.11 | 5.97 | 19.44 | 19.44 | 6 |
| UCI_48 | 14.31 | **14.27** | 14.88 | 33.58 | 33.63 | 33.55 | 14.52 |
| UCI_49 | **9.17** | 9.26 | 9.52 | 9.28 | 26.85 | 10.28 | 9.38 |
| UCI_50 | **12.4** | 12.56 | 12.41 | 13.39 | 22.93 | 15.04 | 12.58 |

Table 14: PinballLoss comparison without temperature-calendar variable feature engineering (I).

| Dataset | | | | | | | | Methods | | | | | | |
|---|---|---|---|---|---|---|---|---|---|---|---|---|---|---|
| | BCEP | CE | KNNR | RFR | SRFR | ERT | SERT | FFNN | LSTM | CNN | LSTNet | WaveNet | N-BEATS | Transformer |
| GEF12_1 | 882.79 | 831.6 | 809.8 | 686.68 | 687.88 | **680.09** | 680.57 | 702.53 | 683.45 | 804.71 | 848.73 | 1641.21 | 994.08 | 819.48 |
| GEF12_2 | 5391.21 | 4689.94 | 4603.71 | 3833.62 | 3841.02 | 3836.54 | 3839.79 | 3838.6 | **3782.73** | 4824.55 | 4815.41 | 9834.17 | 5692.54 | 4780.07 |
| GEF12_3 | 5817.13 | 5060.46 | 4967.21 | 4130.81 | 4138.46 | 4158.01 | 4161.63 | 4141.54 | **4069.16** | 5191.55 | 5198.52 | 10611.12 | 6027.58 | 5337.3 |
| GEF12_4 | 16.74 | 14.75 | 14.72 | 12.98 | 12.99 | 12.87 | **12.86** | 13.1 | 13.01 | 14.69 | 14.75 | 14.8 | 33.27 | 14.85 |
| GEF12_5 | 356.52 | 327.22 | 302.65 | **244.48** | 244.67 | 245.85 | 245.75 | 248.75 | 249.73 | 323.29 | 324.26 | 334.8 | 369.07 | 327.81 |
| GEF12_6 | 5636.84 | 4961.67 | 4838.77 | 4024.48 | 4031.43 | 4053.26 | 4056.44 | 4022.64 | **3959.89** | 5044.73 | 4998.5 | 10422.66 | 5781.23 | 5049.54 |
| GEF12_7 | 5817.13 | 5060.46 | 4967.21 | 4130.81 | 4138.46 | 4158.01 | 4161.63 | 4141.54 | **4069.16** | 5191.55 | 5198.52 | 10611.12 | 6027.58 | 5337.3 |
| GEF12_8 | 150.66 | 138.06 | 143.97 | 123.8 | 124.04 | 121.75 | 121.87 | 124.55 | **119.98** | 138.1 | 140.25 | 322.55 | 322.58 | 140.92 |
| GEF12_9 | 5416.44 | 4618.3 | 4691.03 | **4150.93** | 4152.87 | 4167.93 | 4168.92 | 4346.4 | 4365.51 | 4443.25 | 4658.06 | 4869.88 | 5200.39 | 4493.36 |
| GEF12_10 | 2352.38 | **2160.21** | 7967.5 | 9834.3 | 9835.7 | 9223.33 | 9262.13 | 5984.29 | 5865.75 | 8958.3 | 4869.23 | 7227.35 | 13377.83 | 8780.77 |
| GEF12_11 | 4499.19 | 4175.72 | 3690.62 | 2627.3 | 2633.16 | 2596.65 | **2596.46** | 2939.71 | 2656.91 | 4090.46 | 4266.63 | 8447.85 | 4891.5 | 4049.52 |
| GEF12_12 | 6259.77 | 5817.82 | 5085.18 | **3470.19** | 3476.17 | 3493.22 | 3492.46 | 3666.86 | 3590.75 | 5625.04 | 5673.44 | 11347.51 | 6760.46 | 5707.48 |
| GEF12_13 | 729.03 | 671.94 | 693.42 | 582.43 | 583.39 | **574.14** | 574.49 | 584.34 | 580.09 | 663.49 | 677.34 | 686.45 | 770.82 | 682.34 |
| GEF12_14 | 1275.08 | 1181.48 | 1079.63 | 843.39 | 845.06 | **839.76** | 840.45 | 863.35 | 886.16 | 1147.92 | 1170.62 | 2063.91 | 2065.36 | 1168.65 |
| GEF12_15 | 2479.91 | 2367.5 | 2383.49 | 1987.33 | 1991.64 | **1945.94** | 1947.94 | 1977.17 | 2031.35 | 2310.94 | 2344.5 | 4499.06 | 2815.18 | 2365.66 |
| GEF12_16 | 1629.23 | 1481.79 | 1353.54 | **936.78** | 938.27 | 938.5 | 938.68 | 967.25 | 966.35 | 1452.98 | 1528.32 | 2797.11 | 1765.04 | 1481.21 |
| GEF12_17 | 1333.39 | 1217.72 | 1122.98 | 905.7 | 907.62 | **899.1** | 899.87 | 920.18 | 908.97 | 1187.26 | 1229.87 | 2240.27 | 1420 | 1206.94 |
| GEF12_18 | 10033.59 | 9398.41 | 8858.85 | 6958.4 | 6972.98 | **6886.9** | 6892.25 | 7244.96 | 7066.79 | 9132.19 | 9238.37 | 19009.57 | 10810.68 | 9308.23 |
| GEF12_19 | 4073.7 | 3804.97 | 3711.07 | 3022.4 | 3029.26 | **2970.08** | 2973.39 | 3038.81 | 3066.99 | 3677.03 | 3738.17 | 7466.08 | 4408.83 | 3746.88 |
| GEF12_20 | 2803.65 | 2559.96 | 2658.21 | 2275.06 | 2279.2 | **2241.39** | 2243.34 | 2316.42 | 2336.28 | 2554.55 | 5771.01 | 5773.99 | 2914.96 | 2580.16 |
| GEF14_load | 57.42 | 46.62 | 44.71 | 34.19 | 34.22 | 34.08 | **34.07** | 36.18 | 36.09 | 45.77 | 134.93 | 48.02 | 52.5 | 58.56 |
| GEF17_CT | 82.43 | 74.25 | 74.72 | **50.24** | 50.3 | 50.28 | 50.24 | 51.24 | 55.44 | 71.94 | 71.01 | 171.73 | 88.17 | 72.21 |
| GEF17_ME | 21.63 | 18.57 | 19.88 | 15.95 | 15.97 | 16.09 | 16.1 | **15.57** | 15.7 | 19.59 | 19.77 | 20.81 | 24.45 | 19.8 |
| GEF17_NEMASSBOST | 68.12 | 59.46 | 58.26 | 39.46 | 39.51 | 38.92 | **38.89** | 40.77 | 42.92 | 56.49 | 57.66 | 131.2 | 62.81 | 59.15 |
| GEF17_NH | 29.33 | 24.75 | 25.44 | **18.26** | 18.29 | 18.53 | 18.53 | 18.78 | 18.51 | 24.12 | 24.79 | 66.91 | 66.96 | 26.26 |
| GEF17_RI | 21.24 | 18.96 | 19.41 | 12.75 | 12.76 | 12.66 | 12.64 | **12.34** | 12.92 | 18.22 | 17.69 | 43.34 | 23.46 | 18.16 |
| GEF17_SEMASS | 41.76 | 38.73 | 40.32 | **28.98** | 29.01 | 29.21 | 29.2 | 29.5 | 30.51 | 37.6 | 38.41 | 87.18 | 49.51 | 37.44 |
| GEF17_VT | 15.22 | 12.82 | 13.2 | 11.27 | 11.29 | **10.96** | 10.96 | 11.41 | 11.7 | 13.09 | 13.42 | 28.99 | 28.94 | 13.72 |
| GEF17_WCMASS | 45.71 | 38.6 | 38.61 | **26.38** | 26.41 | 27.16 | 27.15 | 27.47 | 27.8 | 39 | 37.9 | 89.4 | 89.44 | 39.94 |
| Covid19_load | 13847.17 | 11362.8 | 11831.59 | 8211.35 | 8227.93 | **8011.24** | 8017.1 | 8970.64 | 9002.39 | 8375.83 | 8557.02 | 8822.23 | 10083.83 | 8823.82 |
| Bull_Amalia | 11.02 | 9.22 | 9.62 | 9.6 | 9.61 | 9.04 | 9.04 | **8.92** | 8.95 | 9.69 | 9.69 | 13.1 | 13.11 | 9.76 |
| Bull_Goldie | 2.17 | **1.79** | 2.11 | 1.98 | 1.99 | 1.91 | 1.91 | 2.04 | 1.86 | 2 | 2.21 | 2.65 | 2.66 | 2.04 |
| Bull_Lance | 9.76 | 9.52 | 10.51 | 9.92 | 9.95 | 9.58 | 9.59 | 9.59 | 9.77 | **9.33** | 15.91 | 9.4 | 12.14 | 10.05 |
| Bull_Maren | 10.98 | 10.17 | 10.87 | 9.83 | 9.84 | 9.64 | 9.65 | **9.53** | 9.54 | 9.91 | 9.87 | 17.63 | 10.41 | 10.37 |
| Bull_Nathanial | **7.61** | 9.53 | 9.92 | 8.21 | 8.23 | 8.26 | 8.29 | 7.93 | 9.33 | 8.42 | 9.91 | 13.44 | 11.91 | 9.07 |
| Bull_Vanessa | 6.99 | 5.85 | 5.63 | 5.38 | 5.38 | **5.3** | 5.3 | 5.36 | 5.36 | 5.84 | 5.92 | 6.99 | 6.99 | 6.15 |
| Bull_Arthur | 4.24 | 3.9 | 3.88 | 3.69 | 3.7 | 3.65 | 3.65 | **3.62** | 3.64 | 3.7 | 3.85 | 3.87 | 4.19 | 4 |
| Bull_Bernice | 7.15 | 5.45 | 6.06 | 4.38 | 4.38 | **4.27** | 4.28 | 4.74 | 4.74 | 6.15 | 6.29 | 8.87 | 7.47 | 6.9 |
| Bull_Brain | 5.63 | **4.76** | 5.71 | 5.2 | 5.22 | 5.1 | 5.11 | 5.44 | 5.64 | 5.1 | 5.32 | 8.31 | 5.83 | 5.23 |
| Bull_Brenda | 3.89 | 3.12 | 3.32 | 2.9 | 2.9 | **2.81** | 2.81 | 3.43 | 3.15 | 3.37 | 3.48 | 3.94 | 4.69 | 3.46 |
| Bull_Dan | 6.55 | 5.85 | 6.31 | 6.37 | 6.39 | 5.81 | 5.82 | 5.81 | **5.75** | 6.01 | 6.2 | 8.74 | 7.21 | 7 |
| Bull_Dora | 2.52 | 2.23 | 2.07 | **1.96** | 1.96 | 1.98 | 1.99 | 2 | 1.99 | 2.07 | 2.17 | 2.24 | 2.7 | 2.14 |
| Bull_Dottie | 16.54 | 18.77 | 20.01 | 26.86 | 27.01 | 23.25 | 23.26 | 19.7 | 30.76 | 17.11 | 18.68 | 49.43 | 49.68 | **15.6** |
| Bull_Elva | 2 | 1.68 | 1.65 | **1.48** | 1.48 | 1.5 | 1.5 | 1.63 | 1.5 | 1.67 | 1.74 | 1.84 | 1.85 | 1.66 |
| Bull_Gregory | 0.85 | 1.7 | 1.68 | 1.06 | 1.08 | 1 | 0.99 | 2.19 | 5.6 | 0.88 | 0.96 | 11.19 | 0.97 | **0.79** |
| Bull_Jae | 2.41 | 1.97 | 2.35 | 1.91 | 1.91 | 1.87 | 1.87 | **1.84** | 1.95 | 2.12 | 2.15 | 3.21 | 3.22 | 2.22 |
| Bull_Joseph | 3.76 | 2.96 | 3.11 | 2.83 | 2.83 | **2.74** | 2.74 | 2.74 | 2.75 | 3.22 | 3.24 | 4.34 | 4.34 | 3.55 |

Table 15: PinballLoss comparison without temperature-calendar variable feature engineering (II).

| Dataset | | | | | | Methods | | | | | | | | |
|---|---|---|---|---|---|---|---|---|---|---|---|---|---|---|
| | BCEP | CE | KNNR | RFR | SRFR | ERT | SERT | FFNN | LSTM | CNN | LSTNet | WaveNet | N-BEATS | Transformer |
| Bull_Kendra | 5.93 | 5.39 | 5.9 | 5.35 | 5.36 | 5.33 | 5.34 | 5.44 | **5.31** | 5.54 | 5.86 | 11.18 | 6.31 | 5.77 |
| Bull_Krista | 5.99 | 5.67 | 5.47 | 4.75 | 4.76 | **4.74** | 4.74 | 5.34 | 5.45 | 5.99 | 7.11 | 10.28 | 10.3 | 5.96 |
| Bull_Lenny | 1.73 | 1.51 | 1.71 | 1.53 | 1.54 | **1.49** | 1.49 | 1.92 | 1.57 | 1.58 | 2.74 | 2.8 | 2.75 | 1.76 |
| Bull_Linnie | 2.96 | 2.51 | 2.64 | 2.53 | 2.53 | **2.44** | 2.44 | 2.46 | 2.6 | 2.48 | 2.51 | 3.4 | 2.71 | 2.52 |
| Bull_Luke | 7.17 | 5.85 | 5.87 | 5.54 | 5.54 | **5.53** | 5.53 | 5.72 | 5.71 | 5.84 | 6.06 | 6.12 | 6.28 | 5.86 |
| Bull_Magaret | 3.25 | 2.84 | 2.84 | 2.97 | 2.98 | 2.92 | 2.93 | 2.98 | 3.04 | 2.89 | 3.01 | 4.56 | 3.35 | **2.8** |
| Bull_Mervin | 11.21 | 12 | 13.23 | 10.93 | 10.98 | **10.72** | 10.76 | 11.26 | 11.09 | 10.98 | 20.48 | 20.48 | 20.64 | 13 |
| Bull_Miranda | 6.48 | 5.44 | 5.69 | 5.46 | 5.47 | 5.52 | 5.52 | 5.68 | 5.38 | 5.39 | 5.68 | 7.05 | 6.08 | 5.37 |
| Bull_Patrina | 1.31 | 1.11 | 1.1 | 1.07 | 1.07 | **1.06** | 1.06 | 1.11 | 1.12 | 1.09 | 1.19 | 1.19 | 1.19 | 1.11 |
| Bull_Racheal | 14.63 | 11.81 | 12.98 | 8.67 | 8.68 | **8.59** | 8.59 | 9.65 | 8.84 | 13.35 | 14.36 | 20.36 | 20.44 | 13.99 |
| Bull_Reina | 3.14 | 2.75 | 2.58 | 2.28 | 2.29 | **2.26** | 2.27 | 2.53 | 2.61 | 2.84 | 3.23 | 4.53 | 3.71 | 3.26 |
| Bull_Roland | 7.63 | 6.08 | 5.58 | **4.41** | 4.41 | 4.43 | 4.45 | 4.63 | 4.62 | 6.2 | 6.41 | 7.57 | 7.04 | 6.41 |
| Bull_Roseann | 12.46 | 10.31 | 9.58 | 8.92 | 8.93 | **8.87** | 8.87 | 9.47 | 9.09 | 10.12 | 10.38 | 10.42 | 12.34 | 10.39 |
| Bull_Carie | 1.73 | 1.49 | 1.52 | 1.56 | 1.56 | 1.42 | **1.41** | 1.82 | 1.96 | 1.71 | 1.81 | 2.58 | 2.58 | 1.69 |
| Bull_Hugo | 3.51 | 2.84 | 2.9 | 3.44 | 3.44 | 3.34 | 3.35 | 2.76 | **2.74** | 2.9 | 3.04 | 3.05 | 3.07 | 2.93 |
| Bull_Jeremiah | 1.34 | **1.14** | 1.23 | 1.18 | 1.18 | 1.18 | 1.18 | 1.33 | 1.21 | 1.35 | 1.76 | 2.59 | 1.89 | 1.4 |
| Bull_Lettie | 1.24 | 1.04 | 1.04 | **0.97** | 0.97 | 0.98 | 0.98 | 1.04 | 1.03 | 1.01 | 1.12 | 1.28 | 1.26 | 1.08 |
| Bull_Melissa | 6.65 | 5.22 | 5.75 | **4.58** | 4.59 | 4.58 | 4.59 | 4.72 | 4.77 | 5.98 | 5.93 | 8.26 | 8.28 | 6.18 |
| Bull_Perry | 1.99 | 1.61 | 1.54 | 1.35 | 1.35 | **1.34** | 1.34 | 1.38 | 1.37 | 1.64 | 1.71 | 1.72 | 2.17 | 1.69 |
| Bull_Terence | 1.38 | 1.24 | 1.47 | 1.32 | 1.32 | 1.25 | 1.25 | 1.28 | 1.24 | 1.23 | 1.23 | 2.05 | 1.33 | **1.2** |
| Bull_Travis | 14.9 | 12.75 | 14.05 | 12.62 | 12.65 | 11.45 | 11.44 | 11.5 | **11.07** | 14.05 | 14.17 | 18.71 | 29.16 | 16.54 |
| Bull_Chantel | 3.66 | 4.1 | 3.28 | 4.88 | 4.89 | 4.89 | 4.9 | 4.63 | 5 | 3.17 | 3.37 | 5.89 | 5.91 | **3.15** |
| Bull_Yvonne | 2.96 | 2.7 | 3.23 | 2.59 | 2.59 | **2.58** | 2.58 | 3.32 | 3.1 | 2.84 | 2.98 | 5.95 | 5.99 | 2.89 |
| Bull_Hyun | 5.07 | 4.37 | 4.53 | 4.19 | 4.19 | 3.87 | 3.87 | 4.3 | **3.76** | 4.39 | 4.62 | 4.95 | 7.04 | 4.68 |
| Hog_Haywood | 29.5 | 23.35 | 31.62 | 14.7 | 14.75 | **13.52** | 13.54 | 22.82 | 23.48 | 25.34 | 29.21 | 48.08 | 47.87 | 27.48 |
| Hog_Jewel | 27.34 | **23.71** | 41.9 | 33.73 | 33.91 | 29.35 | 29.43 | 27.8 | 39.62 | 26.78 | 64.18 | 64.04 | 33.06 | 35.83 |
| Hog_Leandro | 48.12 | 46.31 | 41.71 | 55.09 | 55.38 | **41.37** | 41.49 | 41.59 | 42.16 | 43.39 | 47.82 | 52.75 | 135.57 | 49.87 |
| Hog_Luvenia | 76.84 | 57.27 | 69.76 | **43.69** | 43.75 | 44.56 | 44.57 | 49.03 | 46.87 | 66.46 | 70.8 | 97.61 | 78.94 | 74.26 |
| Hog_Sonia | 309.32 | 194.12 | 285.98 | **142.78** | 143.12 | 149.8 | 149.92 | 155.7 | 170.05 | 284.13 | 289.36 | 321.1 | 549.81 | 343.58 |
| Hog_Shanti | 154.57 | 191.36 | 199.24 | 170.25 | 170.09 | 169.17 | 168.96 | **136.03** | 150.18 | 155.8 | 171.41 | 372.71 | 372.28 | 158.28 |
| Hog_Betsy | 21.33 | 18.67 | 23.33 | 15.74 | 15.77 | 15.09 | **15.08** | 16.26 | 18.01 | 19.12 | 45.39 | 46.23 | 45.47 | 20.69 |
| Hog_Byron | 341.86 | 322.83 | 421.84 | 271.74 | 272.27 | 244.02 | **243.22** | 264.32 | 320.55 | 321.08 | 350.63 | 389.05 | 432.47 | 364.99 |
| Hog_Candi | 291.15 | 292.27 | 322.53 | 224.92 | 224.81 | 228.42 | 227.86 | **217.16** | 239.73 | 290.05 | 324.25 | 645.56 | 646.74 | 305.03 |
| Hog_Charla | 30.11 | 23.08 | 29.1 | 14.42 | 14.48 | **13.56** | 13.57 | 16.62 | 17.68 | 26.45 | 29.2 | 69.4 | 68.71 | 29.64 |
| Hog_Corey | 34.42 | 17.49 | 16.18 | 12.66 | 12.69 | 12.25 | **12.24** | 14.09 | 14.25 | 15.95 | 16.89 | 28.18 | 18.44 | 18.46 |
| Hog_Cornell | 801.5 | 791.99 | 912.23 | 757.16 | 758.46 | 706.8 | **705.89** | 770.89 | 822.21 | 784.29 | 812.57 | 945.43 | 957.07 | 798.79 |
| Hog_Elizbeth | 25.95 | 22.2 | 32.98 | 17.01 | 17.08 | 17.02 | 17.09 | **16.29** | 18.32 | 25.02 | 25.01 | 64.53 | 64.64 | 24.37 |
| Hog_Elnora | 46.73 | 38.86 | 52.7 | 35.94 | 35.98 | 35.47 | 35.49 | 33.01 | **32.9** | 40.19 | 42.71 | 98.5 | 98.64 | 43 |
| Hog_Leanne | 32.82 | **31.83** | 45.41 | 44.17 | 44.3 | 41 | 41.14 | 41.2 | 33.86 | 41.55 | 69.46 | 68.55 | 69.03 | 46.68 |
| Hog_Nia | 1272.45 | 1205.98 | 1399.11 | 1168.12 | 1173.71 | 1053.79 | **1052.55** | 1079.62 | 1255.2 | 1213.4 | 1346.26 | 1961.3 | 1472.29 | 1217.33 |
| Hog_Richelle | 59.28 | 42.82 | 55.28 | **32.06** | 32.08 | 33.29 | 33.2 | 34.8 | 34.09 | 53.21 | 55.08 | 61.12 | 80.65 | 60.35 |
| Hog_Roger | 18.24 | 15.09 | 18.47 | 14.57 | 14.61 | 14.95 | 14.97 | **13.45** | 18.63 | 17.2 | 18.19 | 40.15 | 24 | 17.64 |
| Hog_Rolando | 55.35 | 40.45 | 53 | 23.49 | 23.55 | **22.51** | 22.52 | 27.42 | 26.88 | 48.79 | 63.06 | 97.15 | 97.01 | 48.57 |
| Hog_Terry | 28.89 | 23.96 | 33.53 | 22.61 | 22.65 | 22.14 | **22.13** | 23.62 | 24.56 | 26.91 | 29.84 | 67.47 | 42.11 | 31.56 |
| Hog_Brad | 33.26 | 26.36 | 31.2 | **19.84** | 19.9 | 20.04 | 20.05 | 21.62 | 19.98 | 30.05 | 31.77 | 85.02 | 85.56 | 39.03 |
| Hog_Crystal | 179.37 | 170.84 | 218.41 | 154.01 | 154.93 | **149.23** | 149.57 | 159.93 | 154.74 | 164.21 | 170.49 | 307.39 | 206.16 | 169.59 |
| Hog_Kevin | 194.64 | 152.94 | 152.34 | 112.82 | **112.8** | 118.45 | 118.3 | 135.06 | 130.94 | 151.88 | 156.89 | 378.66 | 375.61 | 174.55 |
| Hog_Octavia | 186.5 | 135.01 | 188.27 | 119.67 | 119.77 | 124.28 | 124.2 | **118.92** | 127.85 | 173.48 | 177.4 | 353.52 | 225.68 | 188.56 |
| Cockatoo_office_Laila | 28.82 | 17.96 | 22.85 | 14.25 | 14.28 | **14.09** | 14.11 | 15.98 | 14.53 | 23.71 | 24.39 | 41.35 | 41.41 | 24.61 |
| PDB_load | 301.68 | 248.83 | 224.56 | **150.43** | 150.45 | 152.45 | 152.3 | 162.17 | 155.94 | 236.33 | 238.99 | 620.42 | 620.04 | 245.19 |
| Spain_Spain | 949.13 | 733.5 | 576.62 | 453.46 | 454.06 | 441.45 | **441.38** | 535.61 | 535.04 | 588.14 | 606.6 | 1184.82 | 699.68 | 616.4 |

Table 16: PinballLoss comparison with temperature-calendar variable feature engineering (I).

| Dataset | BCEP | CE | KNNR | RFR | SRFR | ERT | SERT | FFNN | LSTM | CNN | LSTNet | WaveNet | N-BEATS | Transformer |
|---|---|---|---|---|---|---|---|---|---|---|---|---|---|---|
| GEF12_1 | 882.79 | 725.94 | 924.99 | 699.95 | 701.29 | 694.73 | 696.53 | 715.31 | 704.28 | 694.97 | 690.12 | 723.47 | **682.71** | 696.92 |
| GEF12_2 | 5391.21 | 3685.88 | 5217.01 | 4009.86 | 4019.72 | 3927.23 | 3938.02 | 3826.33 | 3770.95 | 3721.58 | 3739.57 | 4922.92 | 4937.44 | **3648.04** |
| GEF12_3 | 5817.13 | 3981.61 | 5629.18 | 4324.67 | 4335.41 | 4237.31 | 4249.45 | 4129.25 | 4070.02 | 4028.87 | 4034.83 | 5265.43 | 5321.56 | **3905.57** |
| GEF12_4 | 16.74 | 13.02 | 15.82 | 13.48 | 13.51 | 13.07 | 13.09 | 12.97 | 12.83 | 12.51 | **12.35** | 15.59 | 15.58 | 12.58 |
| GEF12_5 | 356.52 | 240.85 | 342.2 | 259.61 | 259.96 | 246.98 | 247.31 | **233.94** | 240.67 | 243.05 | 242.78 | 283.04 | 287.14 | 245.03 |
| GEF12_6 | 5636.84 | 3857.07 | 5468.26 | 4191.94 | 4201.12 | 4093.91 | 4105.21 | 4022.17 | 4047.47 | 3915.28 | 3958.71 | 5030.4 | 5080.66 | **3821.42** |
| GEF12_7 | 5817.13 | 3981.61 | 5629.18 | 4324.67 | 4335.41 | 4237.31 | 4249.45 | 4129.25 | 4070.02 | 4028.87 | 4034.83 | 5265.43 | 5321.56 | **3905.57** |
| GEF12_8 | 150.66 | 125.8 | 162.84 | 126.6 | 126.88 | 124.88 | 125.19 | 136.57 | 127.81 | 124.59 | 127.58 | 184.78 | 122.05 | **121.27** |
| GEF12_9 | 5416.44 | 4522.87 | 4784.21 | **4313.31** | 4318.72 | 4376.26 | 4386.59 | 4432.2 | 4389.98 | 4377.72 | 4422.99 | 4815.01 | 4779.79 | 4581.38 |
| GEF12_10 | **2352.38** | 8782.93 | 11661.99 | 9777.48 | 9797.6 | 10351.03 | 10373.96 | 8485.51 | 9151.51 | 11022.92 | 8455 | 13706.91 | 7609.93 | 11266.39 |
| GEF12_11 | 4499.19 | 2675.11 | 4692.21 | 2778.62 | 2784.7 | **2659.96** | 2666.19 | 3107.47 | 2787.91 | 2849.07 | 2969.87 | 4612.44 | 4506 | 2842.46 |
| GEF12_12 | 6259.77 | 3508.65 | 6226.82 | 3771.98 | 3777.7 | 3574.22 | 3581.01 | 3675.24 | **3445.59** | 3657.71 | 3797.72 | 5279.55 | 5299.93 | 3671.35 |
| GEF12_13 | 729.03 | 589.98 | 779.87 | 598.01 | 599.31 | 580.82 | 582.19 | 587.06 | 576.65 | **572.5** | 580.79 | 584.51 | 735.5 | 596.2 |
| GEF12_14 | 1275.08 | 834.09 | 1259.24 | 878.26 | 880.18 | 858.12 | 860.45 | **823.71** | 839.54 | 842.62 | 851 | 1046.68 | 1056.14 | 863.4 |
| GEF12_15 | 2479.91 | 1942.29 | 2755.36 | 2025.29 | 2030.33 | 1983.43 | 1989.12 | 1958.72 | 1951.66 | 1941.18 | **1929.21** | 2485.13 | 2046.2 | 1993.46 |
| GEF12_16 | 1629.23 | 917.68 | 1523.33 | 1010.3 | 1012.07 | 966.45 | 968.9 | 918.29 | **910.71** | 940.65 | 959.47 | 1113.59 | 1122.56 | 944.81 |
| GEF12_17 | 1333.39 | 896.93 | 1344.17 | 932.24 | 934.33 | 916.54 | 919.15 | 942.21 | 955.33 | 898.94 | **887.7** | 1268.32 | 946.9 | 918.02 |
| GEF12_18 | 10033.59 | 7016.82 | 10223.66 | 7221.42 | 7237.88 | 7045.5 | 7063.53 | 7302.52 | 7155.24 | **6724.83** | 6994.42 | 9128.22 | 7400.11 | 6954.54 |
| GEF12_19 | 4073.7 | 3030.13 | 4284.08 | 3117.22 | 3124.88 | 3056.9 | 3065.37 | 3073.55 | 2998.16 | **2875.58** | 2997.73 | 3962.97 | 2924.04 | 2958.42 |
| GEF12_20 | 2803.65 | 2411.74 | 3241.79 | 2307.43 | 2312.2 | 2303.38 | 2309.19 | 2497.93 | 2404.11 | 2242.74 | 2343.9 | 3520.92 | 2237.38 | **2227.21** |
| GEF14_load | 57.42 | 27.71 | 44.24 | 34.77 | 34.83 | 31.82 | 31.86 | **25.97** | 27.18 | 29.48 | 28.45 | 35.23 | 30.33 | 32.14 |
| GEF17_CT | 82.43 | 42.78 | 75.63 | 51.96 | 52.04 | 48.73 | 48.81 | **41.68** | 46.4 | 41.93 | 42.81 | 60.33 | 46.77 | 47.35 |
| GEF17_ME | 21.63 | 13.87 | 22.82 | 16.5 | 16.53 | 15.53 | 15.55 | 13.93 | 14.14 | **13.73** | 14.31 | 18.07 | 14.32 | 14.63 |
| GEF17_NEMASSBOST | 68.12 | 37.68 | 57.87 | 40.83 | 40.92 | 39.08 | 39.16 | 37.64 | 39.46 | **34.74** | 36.17 | 46.63 | 42.28 | 40.11 |
| GEF17_NH | 29.33 | **16.18** | 26.41 | 18.86 | 18.89 | 18.03 | 18.06 | 16.53 | 17.58 | 16.94 | 16.82 | 21.6 | 17.59 | 18.15 |
| GEF17_RI | 21.24 | 11.32 | 19.26 | 13.3 | 13.32 | 12.41 | 12.43 | **10.44** | 11.46 | 10.75 | 14.15 | 14.38 | 12.31 | 11.44 |
| GEF17_SEMASS | 41.76 | 27.86 | 40.92 | 30.26 | 30.32 | 28.96 | 29.01 | **26.95** | 28.18 | 27.41 | 27.07 | 35.12 | 28.17 | 30.57 |
| GEF17_VT | 15.22 | 11.41 | 13.48 | 11.72 | 11.74 | 11.07 | 11.09 | 10.34 | 10.58 | **10.28** | 10.36 | 12.73 | 11.07 | 10.78 |
| GEF17_WCMASS | 45.71 | 24.9 | 39.2 | 28.99 | 29.05 | 26.81 | 26.86 | 24.83 | 26.84 | **23.99** | 24.53 | 27.37 | 31.78 | 27.43 |
| Covid19_load | 13847.17 | 18202.22 | 47826.94 | **8248.89** | 8266.91 | 8391.88 | 8407.02 | 22225.41 | 21192.83 | 17190.73 | 20288.67 | 30400.54 | 53663.3 | 21218.05 |
| Bull_Amalia | 11.02 | **9.1** | 10.43 | 9.33 | 9.35 | 9.52 | 9.54 | 9.19 | 9.31 | 9.12 | 9.33 | 9.89 | 9.88 | 9.35 |
| Bull_Goldie | 2.17 | 1.83 | 2.03 | 1.97 | 1.97 | 2.01 | 2.02 | 2.1 | 2.08 | 1.91 | 2.02 | 2.67 | **1.76** | 2.15 |
| Bull_Lance | **9.76** | 10.77 | 13.97 | 9.89 | 9.92 | 10.51 | 10.56 | 9.77 | 10.35 | 10.64 | 10.83 | 17.23 | 9.83 | 10.43 |
| Bull_Maren | 10.98 | 11.22 | 12.2 | **10.05** | 10.08 | 10.18 | 10.21 | 14.31 | 15.79 | 11.29 | 11.56 | 12.33 | 12.68 | 14.26 |
| Bull_Nathanial | **7.61** | 9.38 | 10.02 | 8.77 | 8.8 | 8.68 | 8.72 | 8.92 | 9.24 | 8.56 | 9.77 | 10.63 | 10.72 | 9.26 |
| Bull_Vanessa | 6.99 | 5.76 | 6.05 | 5.41 | 5.41 | **5.38** | 5.39 | 5.54 | 6.11 | 5.46 | 5.52 | 5.58 | 6.18 | 5.49 |
| Bull_Arthur | 4.24 | 4.14 | 4.67 | **3.68** | 3.69 | 3.68 | 3.69 | 4.09 | 4.23 | 3.95 | 3.94 | 5.12 | 3.92 | 4 |
| Bull_Bernice | 7.15 | 4.67 | 5.74 | **4.49** | 4.5 | 4.56 | 4.59 | 4.5 | 4.7 | 4.72 | 4.81 | 5.26 | 5.19 | 5.11 |
| Bull_Brain | 5.63 | 6.13 | 5.51 | **5.28** | 5.3 | 5.93 | 5.97 | 6.12 | 5.8 | 5.67 | 5.67 | 6.31 | 5.44 | 5.3 |
| Bull_Brenda | 3.89 | **2.84** | 3.32 | 2.87 | 2.88 | 2.91 | 2.92 | 2.9 | 2.9 | 2.98 | 3.03 | 3.61 | 3.02 | 3.48 |
| Bull_Dan | 6.55 | 5.86 | 7.24 | 6.28 | 6.3 | 6.14 | 6.17 | 6.28 | 6.23 | **5.75** | 6.01 | 8.35 | 8.1 | 6.6 |
| Bull_Dora | 2.52 | 2.18 | 2.14 | **2.03** | 2.03 | 2.11 | 2.13 | 2.06 | 2.03 | 2.07 | 2.11 | 2.29 | 2.29 | 2.12 |
| Bull_Dottie | **16.54** | 20.08 | 30.28 | 26.12 | 26.28 | 26.57 | 26.66 | 28.45 | 32.33 | 20.65 | 21.5 | 21.68 | 39.46 | 19.91 |
| Bull_Elva | 2 | 1.68 | 1.77 | **1.53** | 1.53 | 1.6 | 1.6 | 1.58 | 1.57 | 1.61 | 1.65 | 1.65 | 1.76 | 1.66 |
| Bull_Gregory | 0.85 | 2.2 | 3.34 | **0.84** | 0.84 | 1.02 | 1.01 | 2.77 | 3.99 | 1.57 | 2.22 | 8.88 | 1.97 | 0.98 |
| Bull_Jae | 2.41 | **1.88** | 2.21 | 1.92 | 1.92 | 1.95 | 1.96 | 1.96 | 1.96 | 1.96 | 2.93 | 2.93 | 2.01 | 2.29 |
| Bull_Joseph | 3.76 | **2.85** | 3.29 | 2.97 | 2.98 | 2.86 | 2.87 | 2.89 | 2.97 | 2.96 | 2.95 | 3.96 | 3.09 | 3.16 |

Table 17: PinballLoss comparison with temperature-calendar variable feature engineering (II).

| Dataset | BCEP | CE | KNNR | RFR | SRFR | ERT | SERT | FFNN | LSTM | CNN | LSTNet | WaveNet | N-BEATS | Transformer |
|---|---|---|---|---|---|---|---|---|---|---|---|---|---|---|
| Bull_Kendra | 5.93 | 6.34 | 7.65 | **5.36** | 5.38 | 5.43 | 5.45 | 6.24 | 5.94 | 6.09 | 6.34 | 10.86 | 10.93 | 6.35 |
| Bull_Krista | 5.99 | 6.08 | 6.39 | **4.65** | 4.67 | 5.33 | 5.35 | 6.16 | 6.31 | 6.1 | 6.68 | 8.05 | 6.38 | 6.68 |
| Bull_Lenny | 1.73 | 1.67 | 1.58 | 1.46 | 1.47 | 1.52 | 1.52 | 2.72 | 1.81 | **1.44** | 1.53 | 2.71 | 2.71 | 1.53 |
| Bull_Linnie | 2.96 | 2.65 | 2.68 | **2.49** | 2.5 | 2.59 | 2.6 | 2.84 | 2.79 | 2.68 | 2.94 | 3.87 | 3.82 | 2.52 |
| Bull_Luke | 7.17 | 5.89 | 6.16 | **5.52** | 5.53 | 5.79 | 5.81 | 6.08 | 6.02 | 5.82 | 6.02 | 6.41 | 6.07 | 5.93 |
| Bull_Magaret | 3.25 | 2.97 | 3.23 | 2.95 | 2.96 | 3.21 | 3.22 | 3.54 | 3.71 | 2.96 | 2.97 | 4.38 | 4.32 | 3.11 |
| Bull_Mervin | 11.21 | 12.26 | 11.51 | 11.2 | 11.26 | 11.19 | 11.24 | 11.74 | 12.25 | 10.96 | 11 | **10.72** | 15.32 | 11.81 |
| Bull_Miranda | 6.48 | 5.86 | 6 | 5.68 | 5.69 | 6.28 | 6.31 | 6.31 | 6.26 | 5.96 | 6.22 | 7.86 | 5.83 | 5.83 |
| Bull_Patrina | 1.31 | 1.08 | 1.11 | **1.06** | 1.06 | 1.09 | 1.09 | 1.1 | 1.17 | 1.11 | 1.11 | 1.21 | 1.21 | 1.18 |
| Bull_Racheal | 14.63 | 9.13 | 11.55 | **8.75** | 8.77 | 8.77 | 8.8 | 9.01 | 9.18 | 9.81 | 9.95 | 9.94 | 10.74 | 13.03 |
| Bull_Reina | 3.14 | 2.58 | 2.82 | **2.36** | 2.37 | 2.5 | 2.52 | 2.9 | 2.74 | 2.63 | 2.73 | 3.22 | 2.64 | 2.92 |
| Bull_Roland | 7.63 | 4.95 | 5.44 | **4.5** | 4.51 | 4.79 | 4.83 | 4.62 | 4.53 | 4.99 | 4.8 | 4.99 | 5.22 | 5.94 |
| Bull_Roseann | 12.46 | 9.87 | 10.01 | **8.91** | 8.92 | 9.37 | 9.41 | 9.5 | 9.58 | 9.61 | 9.68 | 10.04 | 9.67 | 9.53 |
| Bull_Carie | 1.73 | 1.68 | 2.06 | 1.58 | 1.59 | **1.53** | 1.54 | 2.19 | 2.38 | 1.84 | 1.94 | 2.04 | 1.86 | 1.93 |
| Bull_Hugo | 3.51 | 3.23 | 3.32 | 3.31 | 3.32 | 3.6 | 3.61 | 3.15 | 2.97 | 3 | 3.02 | 2.99 | 3.31 | **2.87** |
| Bull_Jeremiah | 1.34 | **1.1** | 1.71 | 1.15 | 1.16 | 1.2 | 1.2 | 1.45 | 1.54 | 1.28 | 1.61 | 1.94 | 2.39 | 1.41 |
| Bull_Lettie | 1.24 | 1.02 | 0.99 | **0.97** | 0.97 | 1 | 1.01 | 1.04 | 1.06 | 1.03 | 1.05 | 1.16 | 1.12 | 1.04 |
| Bull_Melissa | 6.65 | 4.6 | 5.67 | 4.69 | 4.7 | 4.69 | 4.71 | **4.56** | 4.68 | 5.19 | 4.64 | 4.75 | 5.85 | 5.8 |
| Bull_Perry | 1.99 | 1.41 | 1.52 | **1.33** | 1.33 | 1.41 | 1.43 | 1.38 | 1.34 | 1.4 | 1.42 | 1.42 | 1.46 | 1.65 |
| Bull_Terence | 1.38 | 1.29 | 1.98 | 1.35 | 1.35 | 1.34 | 1.34 | 1.48 | 1.54 | 1.28 | 1.29 | 2.1 | 2.08 | **1.27** |
| Bull_Travis | 14.9 | 12.16 | 17.93 | **11.34** | 11.42 | 11.49 | 11.52 | 12.58 | 13.83 | 12.15 | 23.02 | 24.39 | 24.23 | 16.16 |
| Bull_Chantel | 3.66 | 4.63 | 5.26 | 4.95 | 4.97 | 5.57 | 5.6 | 5.46 | 5.58 | 4.78 | 4.95 | 6.28 | 6.25 | 4.2 |
| Bull_Yvonne | 2.96 | 2.66 | 3.53 | **2.55** | 2.56 | 2.57 | 2.58 | 3.66 | 3.92 | 2.71 | 2.94 | 6.05 | 3.63 | 2.92 |
| Bull_Hyun | 5.07 | **3.82** | 5.2 | 3.99 | 4 | 4.32 | 4.35 | 3.85 | 3.88 | 4.1 | 4.18 | 4.33 | 4.3 | 4.27 |
| Hog_Haywood | 29.5 | 17.5 | 29.59 | 14 | 14.06 | **13.56** | 13.63 | 23.61 | 26.58 | 22.57 | 26.83 | 29.19 | 26.8 | 24.64 |
| Hog_Jewel | **27.34** | 29.29 | 32.95 | 28.34 | 28.45 | 28.15 | 28.26 | 37.4 | 40.15 | 29.31 | 30.57 | 41.53 | 41.51 | 40.97 |
| Hog_Leandro | 48.12 | 55.65 | 83.09 | 42.1 | 42.29 | **40.94** | 41.13 | 53.8 | 56.32 | 44.6 | 52.9 | 77.6 | 77.96 | 49.23 |
| Hog_Luvenia | 76.84 | 49.78 | 66.64 | 46.41 | 46.52 | **45.14** | 45.26 | 48.37 | 48.35 | 58.12 | 55.58 | 64.65 | 64.62 | 64.9 |
| Hog_Sonia | 309.32 | 202.72 | 266.93 | **168.39** | 169 | 170.43 | 171.2 | 194.9 | 195.13 | 201.2 | 206.19 | 217.3 | 208.54 | 214.78 |
| Hog_Shanti | **154.57** | 247.07 | 240.88 | 167.87 | 168.63 | 178.76 | 180.29 | 207.91 | 226.18 | 172.04 | 197.76 | 333.64 | 333.89 | 171.55 |
| Hog_Betsy | 21.33 | 16.56 | 22.16 | 16.64 | 16.68 | 15.86 | 15.89 | **15.79** | 16.94 | 16.75 | 17.04 | 20.16 | 17.75 | 18.96 |
| Hog_Byron | 341.86 | 510.98 | 422.65 | 235.95 | 236.54 | **225.91** | 225.95 | 468.27 | 481.04 | 451.13 | 474.56 | 797.66 | 802.84 | 463.98 |
| Hog_Candi | 291.15 | 236.98 | 428.92 | **230.67** | 230.82 | 248.99 | 249.3 | 252.59 | 249.22 | 271.45 | 285.27 | 415.88 | 276 | 286.38 |
| Hog_Charla | 30.11 | 14.7 | 31.48 | 15.31 | 15.35 | **14.46** | 14.5 | 15.68 | 17.31 | 20.48 | 20.99 | 23.85 | 24.75 | 20.48 |
| Hog_Corey | 34.42 | 16.85 | 13.88 | **12.88** | 12.92 | 13.37 | 13.42 | 14.59 | 14.37 | 14.08 | 14.45 | 16.62 | 13.58 | 14.29 |
| Hog_Cornell | 801.5 | 899.75 | 1326.41 | **727.64** | 729.27 | 774.91 | 776.91 | 1101.52 | 1126.15 | 878.17 | 1083.78 | 1348.96 | 1356.9 | 827.11 |
| Hog_Elizbeth | 25.95 | 31.27 | 32.32 | 16.49 | 16.56 | **16.36** | 16.42 | 17.95 | 22.16 | 24.85 | 28.05 | 34.2 | 35.1 | 21.19 |
| Hog_Elnora | 46.73 | 54.45 | 50.33 | 35.1 | 35.17 | 38.22 | 38.32 | 37.16 | 37.26 | **33.26** | 37.97 | 47.93 | 37.57 | 35.97 |
| Hog_Leanne | 32.82 | 42.67 | 48.12 | 42.84 | 43.01 | 41.76 | 41.91 | 47.17 | 44.12 | **32.03** | 36.71 | 49.19 | 46.17 | 41.56 |
| Hog_Nia | 1272.45 | 1327.4 | 1870.7 | **1082.83** | 1085.92 | 1174.08 | 1177.99 | 1461.89 | 1474.52 | 1337.24 | 1498.89 | 1887.36 | 1890.6 | 1243.84 |
| Hog_Richelle | 59.28 | 38.2 | 60.21 | 38.68 | 38.76 | 36.75 | 36.85 | **36.32** | 38.99 | 42.08 | 41.36 | 54.4 | 46.82 | 40.37 |
| Hog_Roger | 18.24 | **13.78** | 25.78 | 15.26 | 15.32 | 17.05 | 17.13 | 14.89 | 16.95 | 15.08 | 15.05 | 28.05 | 20.08 | 15.82 |
| Hog_Rolando | 55.35 | 26.46 | 55.16 | **22.92** | 22.99 | 23.03 | 23.12 | 25.83 | 26.67 | 42.05 | 41 | 42.98 | 42.87 | 41.34 |
| Hog_Terry | 28.89 | 24.91 | 34.21 | 23.78 | 23.85 | **22.85** | 22.91 | 23.36 | 24.59 | 23.74 | 24.08 | 45.05 | 44.89 | 31.16 |
| Hog_Brad | 33.26 | 22.34 | 37.19 | 22 | 22.07 | 21.19 | 21.26 | **17.97** | 18.61 | 22.52 | 21.08 | 31.88 | 31.81 | 23.3 |
| Hog_Crystal | 179.37 | 187.2 | 190.28 | 145.19 | 145.5 | **139.56** | 139.7 | 335.42 | 352.92 | 188.18 | 199.5 | 449.99 | 203.65 | 175.33 |
| Hog_Kevin | 194.64 | 122.35 | 145.08 | 124.14 | 124.34 | **120.24** | 120.44 | 132.76 | 138.54 | 124.69 | 127.48 | 160.99 | 163.14 | 158.67 |
| Hog_Octavia | 186.5 | **112.02** | 178.31 | 132.33 | 132.65 | 131.59 | 131.88 | 142.97 | 147.03 | 139.84 | 146.58 | 186.05 | 180.35 | 148.23 |
| Cockatoo_office_Laila | 28.82 | 14.05 | 19.58 | 14.59 | 14.63 | 14.95 | 15.02 | **13.61** | 14.39 | 16.36 | 16.61 | 16.67 | 18.08 | 16.37 |
| PDB_load | 301.68 | 126.17 | 233.53 | 155.93 | 156.07 | 141.14 | 141.17 | **111.09** | 111.62 | 129.62 | 128.64 | 131.84 | 142.72 | 132.61 |
| Spain_Spain | 949.13 | 640.76 | 654.03 | 567.73 | 568.4 | **559.93** | 561.06 | 603.6 | 632.48 | 598.73 | 590.36 | 579.11 | 718.85 | 575.12 |

