# OpenReview forum: "Benchmarks and Custom Package for Electrical Load Forecasting"
_ICLR.cc/2024/Conference — ICLR 2024 Conference Withdrawn Submission_

### Official Review · Reviewer_rLha · 2023-10-31

**Soundness:** 2 fair
**Presentation:** 2 fair
**Contribution:** 3 good
**Rating:** 3
**Confidence:** 3

**Summary:**

This paper proposes a benchmark study of Time Series Electricity Load Demand Forecasting (TSELDF), especially focusing on predicting Electricity Load Demand (ELD) of Building(s). This benchmark is accompanied by a platform, which contains necessary codes for TSELDF, ranging from pre-processing, forecasting models and post-processing. The benchmark notably demonstrates the importance of engineered features and appropriate loss function. The latter is important because a good ELD prediction is not necessarily the closest prediction to the ground truth, but one that take into consideration the cost.

Authors experimented their platform on 11 datasets and against various baselines on two scenarios: Probabilistic forecasting and point forecasting.

**Strengths:**

Authors put a great effort in gathering techniques, models and datasets on TSELDF and in creating a Git repository. This will definitely help future users test their platform and enhance it in the future.

They also thoroughly described the collected dataset and experiment settings, which will help other researchers in reproducing their results.

Finally, they clearly explained the goal of TSELDF and set the scope of the benchmark: not only focusing on time series forecasting, but more importantly on TSELDF, which need different loss function.

**Weaknesses:**

The paper is interesting, but I have a feeling of unfinished. I end up with more questions than answers.

Here, Authors provide a library compiling various tools and show the results of specific configurations of these elements for given datasets. However, the results demonstrated are already “known” (importance of temperature and specific loss function, because extracted from existing paper referenced by Authors). The question now is what is the novelty of Authors apart from compiling the techniques and models? I would expect from a benchmark paper to guide users on when to use specific engineered features or specific techniques depending on the dataset, the experiment setup, etc. And discuss more the results.

In addition, the dataset analysis could be further improved in order to better understand their differences and better comprehend reason why in some configuration a given models should be preferred to another one.

Moreover, the Authors did not address the limitation of their paper such as missing latest SOTA transformers [1] or Linear models [2], applicability to other load forecasting dataset (What does building stand for here? Is it residential building, office or industrial? Or any. Time constraint and temperature importance will vary a lot depending on such information).

Discussion that could be started in the paper:
 * What is the reason for temperature to not perform well with DL models, is it because of the integration is too simplistic or is it inherent to the DL architecture?
 * How will the cost loss function will apply in a large-scale scenario? Or temperature?

Finally, in my opinion the paper could be re-written in order to make content clearer and better convey the ideas. And a proof-read is required cf. next section.

[1] https://openreview.net/pdf?id=cGDAkQo1C0p

[2] https://arxiv.org/pdf/2305.10721.pdf

**Questions:**

Please find below some additional questions or comments I had when reading the paper:

## Questions
> For this reason, calendar variables significantly impact load forecasting at this level.

Claim? Where is the proof? Or known fact Where is the ref?

## Comments
> On the one hand, the load is largely influenced by many external factors, such as temperature or calendar variables

Loads are not really influence by these factors, but it a consequence of these factors pushing users to change their behavior. For instance [3] [4] showed that human dynamics only affect ELD.

> we will provide various related feature engineering in our package and discuss the impact on load forecasting models based on temperature feature engineering.

Authors only demonstrated temperature…

> Because the aggregated-level load results from multiple load aggregations, it typically exhibits more pronounced periodicity and seasonality

Proof? Authors could demonstrate this by plotting ACF of each dataset, providing them in appendix and showing the seasonality difference of each dataset.

> Building-level loads change very dramatically, resulting in significant uncertainty

In my opinion, this sentence contradicts the previous one. Why would an aggregation of individual building level load that are very different will necessarily exhibits more pronounced periodicity? There is a scale parameter to take into consideration, no?

## Suggestions
### Fig 3:
Authors should provide the figures for the other datasets in appendix otherwise it might look like they exhibit only the advantageous results. In addition, for clearer understanding, use color to differentiate models (DL and non-DL models). And perhaps, a log axis will be more suitable.

### Git:

Code might need more comments to better help non-initiated people that would like to start using such a package.

### Appendix:

Figures 19 and 20 are missing some subplots, no?

### Table 3:

Include in the caption on Table 3 the meaning of bold values.


A proof-read is required, cf. the following:
 * “our dataset is collected and the characteristics of the dataset” datasets are collected […] of each dataset
 * “global energy forecasting competitions” why not use the acronym introduce in introduction
 * “7 of the data we collect” we collected
 * “In contrast, the load of the building level, which can also be seen as a part of the aggregated load.” Sentence needs to be revised.
 * “For sequence models like the LSTM” like LSTM

[3] Elucidating the extent by which population staying patterns help improve electricity load demand predictions
G Habault, S Wada, R Kimura, C Ono - 2020 IEEE International Conference on Big Data

[4] Detecting errors in short-term electricity demand forecast using people dynamics
G Habault, Y Nishimura, K Yoshihara, C Ono - 2019 IEEE International Conference on Big Data (Big …, 2019

---

### Official Review · Reviewer_zDou · 2023-11-01

**Soundness:** 2 fair
**Presentation:** 2 fair
**Contribution:** 2 fair
**Rating:** 3
**Confidence:** 4

**Summary:**

The paper focuses on the challenging problem of electrical load forecasting and proposes a comprehensive framework to tackle it effectively. One of the central contributions is the introduction of an asymmetric loss function, designed to capture the varying costs associated with overestimating and underestimating electrical loads. This is a significant deviation from traditional symmetric loss functions commonly used in the literature and aims to provide a more realistic evaluation of forecasting performance.

In addition to the novel loss function, the paper also introduces temperature-based feature engineering to improve the performance of deep learning models for load forecasting. The authors argue that incorporating environmental features like temperature can lead to more accurate and reliable forecasts.

The paper goes beyond mere theoretical contributions by providing an open-source package that is modular and extensible. This package encompasses the entire power forecasting process and allows users to easily integrate different forecasting methods and feature engineering techniques.

For evaluation, a comprehensive set of benchmarks is presented, covering multiple datasets that include both aggregated-level and building-level electrical loads. The datasets originate from various sources, including different forecasting competitions and publicly available repositories. Several metrics, such as Pinball Loss and MAPE (Mean Absolute Percentage Error), are employed to assess the performance comprehensively.

**Strengths:**

Assessment of Strengths
Originality
Asymmetric Loss Function: The introduction of an asymmetric loss function to capture the varying cost implications of underestimating and overestimating electrical load is a noteworthy original contribution. This deviates from conventional symmetric loss functions, offering a potentially more realistic performance assessment for the load forecasting problem.

Comprehensive Feature Engineering: The paper incorporates temperature-based feature engineering, which provides a new angle to improve the performance of load forecasting models. This is a creative combination of environmental data with load data to improve the forecasting ability of deep learning models.

Multi-dataset Benchmarking: The paper uses multiple datasets for evaluation, including those from different competitions and different kinds of loads (aggregated-level and building-level). This contributes to the originality as it goes beyond the single-dataset evaluation commonly seen in the literature.

Quality
Extensive Evaluation: The paper is rigorous in its evaluation methodology, employing multiple metrics and datasets to assess the performance of the proposed methods. This lends credibility to the claims made in the paper.

Open-source Codebase: The provision of an open-source code repository enhances the quality and reproducibility of the work, allowing for validation and extension by the broader research community.

Clarity
Well-Structured: The paper is well-organized, systematically presenting the methodology, feature engineering techniques, evaluation metrics, and results. This makes it easier for readers to follow the research contributions.

Detailed Dataset Description: The comprehensive description of the datasets, including their origin, characteristics, and preprocessing steps, adds to the clarity and makes it easier for other researchers to understand the context and limitations of the study.

Significance
Practical Relevance: Electrical load forecasting is a problem of considerable practical significance. The paper’s focus on capturing the cost implications of forecasting errors through a custom loss function has potential real-world impact.

Extensibility of Framework: The proposed package is described as highly extensible, allowing for modifications and improvements. This could serve as a foundational tool for further research in load forecasting and even in other time series forecasting problems.

Potential for Cross-Domain Application: The techniques used in the paper, especially the asymmetric loss function and feature engineering, have the potential to be applied in other domains of time-series forecasting, broadening the significance of the work.

**Weaknesses:**

Methodological Weaknesses
Choice of Custom Loss Function: The paper introduced an asymmetric loss function to capture the difference in cost due to load underestimation and overestimation. While the authors argue that this asymmetric loss function improves performance, its efficacy is inconsistent across datasets (e.g., GEF17 and Spain). A deeper explanation is needed regarding why it performs poorly on aggregated-level data like GEF17.

Lack of Theoretical Justification: The paper lacks a theoretical foundation that explains why the chosen custom loss function is more suitable than other potential loss functions (e.g., Huber loss, quantile loss).

Non-Uniformity in Dataset Treatment: The paper applies temperature-based feature engineering uniformly across datasets without considering the nature of the data. For instance, the Covid19 dataset, which shows a decline in electrical load due to the pandemic, might require different treatment.

Oversimplification in Feature Engineering: The paper mentions the use of a single temperature series for multiple load series in GEF12. This oversimplification can lead to a loss of model expressiveness and performance degradation.

Issues with Equations
Lack of Generalization: The mathematical equations for the custom loss function are limited to the application domain. A generalized mathematical model would add to the rigor of the research.
Weaknesses in Diagrams and Graphs
Limited Insight: Figures such as Figure 3 and Figure 4 provide limited insight into how the feature engineering process actually affects the model's performance. These figures could be enhanced by including confidence intervals or additional metrics.

Ambiguity in Visualization: The visualizations of load data (e.g., Fig. 12, Fig. 13) lack details on statistical properties like skewness, kurtosis, which are critical in time-series data.

Benchmarking and Comparative Analysis

Inconsistency in Metric Reporting: The paper reports performance in terms of Pinball Loss and cost but doesn't consider other widely-used metrics like RMSE or MAPE for a comprehensive evaluation.

Scalability and Efficiency:

The paper lacks an analysis of the computational complexity of the proposed methods.
There is no discussion on the scalability of the proposed model to larger datasets or more complex systems, which is crucial for real-world applications.

Conclusions and Future Directions
Lack of Rigor in Conclusions: The conclusions make broad claims about the utility and extensibility of the developed package but lack empirical or theoretical support for these claims.

Limited Discussion on Model Robustness: The paper does not discuss the robustness of the proposed model in the face of data anomalies, missing data, and outliers, which are common in real-world applications.

**Questions:**

Methodology and Model Design
Justification for Asymmetric Loss Function: Could you provide more theoretical justification for the choice of an asymmetric loss function over other well-established loss functions like Huber loss or quantile loss? How does it specifically benefit the forecasting model in capturing the cost of errors?

Effectiveness Across Datasets: Why does the custom loss function perform inconsistently across different types of datasets, particularly underperforming on aggregated-level data like GEF17? Could this be an inherent limitation of the loss function?

Feature Engineering Uniformity: Is the temperature-based feature engineering universally beneficial across all datasets? For instance, how does it affect datasets with unique characteristics like the Covid19 dataset?

Evaluation Metrics and Benchmarks
Choice of Evaluation Metrics: Why are Pinball Loss and cost chosen as the primary evaluation metrics? Could you include other widely-used metrics like RMSE or MAPE to provide a more comprehensive evaluation?

Comparison with State-of-the-Art: The paper does not seem to compare the proposed methods with existing state-of-the-art techniques in electrical load forecasting. Could you elaborate on how your method compares with other state-of-the-art models in terms of accuracy and computational efficiency?

Data Handling and Preprocessing
Handling of Missing and Outlier Data: How does the proposed model handle missing values and outliers, particularly in building-level data like the Hog dataset? Could you elaborate on the robustness of the model in such cases?

Data Granularity in Feature Engineering: Could you explain the rationale behind using a single temperature series for multiple load series in the GEF12 dataset? Do you anticipate this affecting the model's performance?

Generalizability and Scalability
Scalability of the Codebase: Could you provide insights into the scalability of the provided code repository? Is it designed to handle large-scale data and is it optimized for computational efficiency?

Temporal Consistency: The paper mentions the use of data from various years. How does the model handle the temporal inconsistencies, and how does this affect the model's generalizability over time?

Miscellaneous
Clarification on Conclusions: The conclusions claim high extensibility and utility of the developed package. Could you provide empirical or theoretical evidence to support this claim?

Further Research Directions: Are there any plans to extend this work to other domains or types of time-series data? What modifications would be necessary for such extensions?

---

### Official Review · Reviewer_wtQT · 2023-11-02

**Soundness:** 3 good
**Presentation:** 3 good
**Contribution:** 2 fair
**Rating:** 5
**Confidence:** 4

**Summary:**

Electrical load forecasting is an important task for power industries and this paper proposes benchmarks and a package for this task. The paper rightly claims that electrical load forecasting is influenced by several external factors such as different weather conditions, days of the calendar, external events, etc. It proposes a customized loss function for this task and included standard time series, ML and DL approaches as baselines on 11 load datasets for both aggregated and individual building level load forecasting. The authors make the source code and the datasets available with the paper.

**Strengths:**

The paper has the following strengths:

1. Electrical load forecasting is an important task and the authors intend to provide benchmarks and a package for this task.
2. They have considered both individual and aggregated load forecasting, and explain the effect of external variables on this task.
3. Authors also have provided a customized loss function which captures the notion of the error of load forecasting to the actual impact of that on the power utilities.
4. Source code and datasets are available.

**Weaknesses:**

Paper has the following weaknesses:

1. Load forecasting is a spatio-temporal prediction problem. Electrical loads at neighboring regions are highly correlated. However, the spatial aspect of the problem is completely missing from the paper. The authors have considered both individual and aggregated forecasting - but have not used the spatial information present in the dataset to improve the forecasting.

2. In Section 1, "Underestimating predictions at peak points..." - It is not clear if the impact is different for over-estimating and under-estimating predictions during peak points? Intuitively, over-estimating during peak points implies that one has to increase the capacity of the power generation which can lead to significant cost for infrastructural changes. Similarly, under-estimating during peak points implies additional purchase of power during the high demand time which may again cost significantly if it happens frequently. This is also related to the demand response problem in smart grid. Please check: "Siano, Pierluigi. "Demand response and smart grids—A survey." Renewable and sustainable energy reviews 30 (2014): 461-478".

3. The datasets used are not large in scale. UCI is the only dataset used where it has both a good number of series and length.

4. In Section 5.1, text part just summarizes the observation in the tables containing the experimental results. However, the reasoning part is missing. For e.g., why do non-deep learning methods perform better than deep learning methods without temperature transformation strategy? Why do some DL methods perform better than other more complicated DL methods?

5. The baseline algorithms used in the paper are very standard time series forecasting or DL algorithms. They are also easily available in many ML/DL packages. It is not clear why authors do not use any existing approaches customized for load forecasting? There are certainly approaches available in electrical load forecasting literature which are tailored for this task. That way the package can be more relevant for researchers working in the domain.

6. Authors have not compared if "aggregation followed by prediction" or "prediction followed by aggregation" or some other type of strategy is preferred for aggregate level forecasting where the corresponding individual level power consumption series are available. Please check "Bandyopadhyay, Sambaran, et al. "Individual and aggregate electrical load forecasting: One for all and all for one." Proceedings of the 2015 ACM Sixth International Conference on Future Energy Systems. 2015".

7. Apart from the datasets and external factors used for this task, the paper looks very similar to any standard forecasting task. Is it possible to add more technical evidence to show how load forecasting is different from a standard time series forecasting problem with dependency on external variables?

**Questions:**

Please see the weaknesses section above.